# Flag Aggregator: Scalable Distributed Training under Failures and Augmented Losses using Convex Optimization

**Hamidreza Almasi**    **Harsh Mishra**    **Balajee Vamanan**    **Sathya N. Ravi**
Department of Computer Science
University of Illinois Chicago
`{halmas3, hmishr3, bvamanan, sathya}@uic.edu`

## Abstract

Modern ML applications increasingly rely on complex deep learning models and large datasets. There has been an exponential growth in the amount of computation needed to train the largest models. Therefore, to scale computation and data, these models are inevitably trained in a distributed manner in clusters of nodes, and their updates are aggregated before being applied to the model. However, a distributed setup is prone to Byzantine failures of individual nodes, components, and software. With data augmentation added to these settings, there is a critical need for robust and efficient aggregation systems. We define the quality of workers as reconstruction ratios $\in (0, 1]$, and formulate aggregation as a Maximum Likelihood Estimation procedure using Beta densities. We show that the Regularized form of log-likelihood wrt subspace can be approximately solved using iterative least squares solver, and provide convergence guarantees using recent Convex Optimization landscape results. Our empirical findings demonstrate that our approach significantly enhances the robustness of state-of-the-art Byzantine resilient aggregators. We evaluate our method in a distributed setup with a parameter server, and show simultaneous improvements in communication efficiency and accuracy across various tasks[1].

## 1 Introduction

**How to Design Aggregators?** We consider the problem of designing aggregation functions that can be written as optimization problems of the form,

$$\mathcal{A}(g_1, \ldots, g_p) \in \arg \min_{Y \in C} A_{g_1, \ldots, g_p}(Y), \tag{1}$$

where $\{g_i\}_{i=1}^p \subseteq \mathbb{R}^n$ are given estimates of an unknown summary statistic used to compute the *Aggregator* $Y^*$. If we choose $A$ to be a quadratic function that decomposes over $g_i$'s, and $C = \mathbb{R}^n$, then we can see $\mathcal{A}$ is simply the standard mean operator. There is a mature literature of studying such functions for various scientific computing applications Grabisch et al. (2009). More recently, from the machine learning standpoint there has been a plethora of work Balakrishnan et al. (2017); Diakonikolas et al. (2019); Cheng et al. (2022); Diakonikolas et al. (2022) on designing provably robust aggregators $\mathcal{A}$ for mean estimation tasks under various technical assumptions on the distribution or moments of $g_i$.

**Distributed ML Use Cases.** Consider training a model with a large dataset such as ImageNet-1K Russakovsky et al. (2015) or its augmented version which would require data to be distributed over $p$ workers and uses back propagation. Indeed, in this case, $g_i$'s are typically the gradients computed by individual workers at each iteration. In settings where the training objective is convex, the convergence and generalization properties of distributed optimization can be achieved by defining $\mathcal{A}$ as a weighted combination of gradients facilitated by a simple consensus matrix, even if some $g_i$'s are noisy Tsianos and Rabbat (2012); Yang et al. (2019a). In a distributed setup, as long as the

---

[1]Our code is available at `https://github.com/hamidralmasi/FlagAggregator`

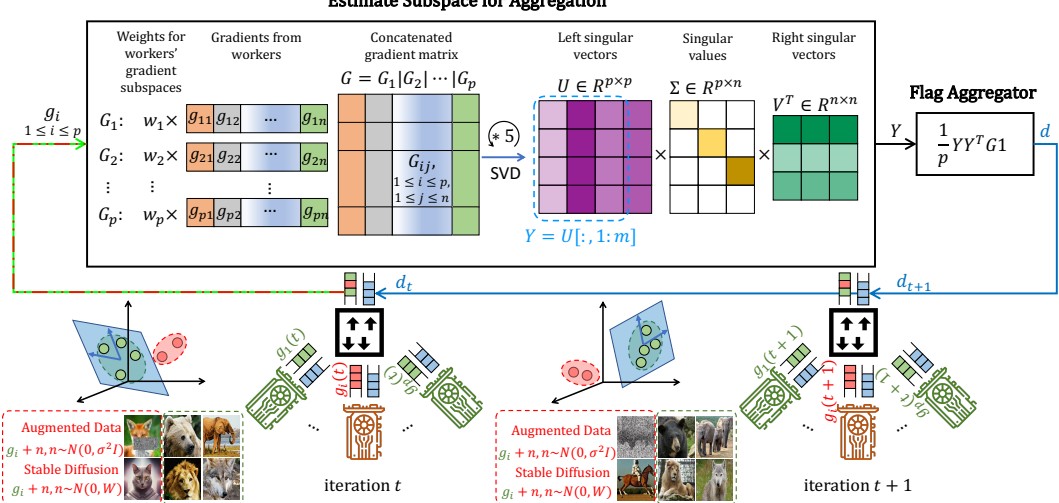

Figure 1: Robust gradient aggregation in our distributed training framework. In our applications, each of the $p$ workers provides gradients computed using a random sample obtained from given training data, derived synthetic data from off-the-shelf Diffusion models, and random noise in each iteration. Our Flag Aggregator (FA) removes high frequency noise components by using few rounds of Singular Value Decomposition of the concatenated Gradient Matrix $G$, and provides new update $Y^*$.

model is convex we can simultaneously minimize the total iteration or communication complexity to a significant extent i.e., it is possible to achieve convergence *and* robustness under technical assumptions on the moments of (unknown) distribution from which $g_i$'s are drawn. However, it is still an open problem to determine the optimality of these procedures in terms of either convergence or robustness Blanchard et al. (2017); Farhadkhani et al. (2022).

**Potential Causes of Noise.** When data is distributed among workers, hardware and software failures in workers Bautista-Gomez et al. (2016); Schroeder and Gibson (2007); Gill et al. (2011) can cause them to send incorrect gradients, which can significantly mislead the model Baruch et al. (2019). To see this, let's consider a simple experiment with 15 workers, that $f$ of them produce uniformly random gradients. Figure 2 shows that the model accuracy is heavily impacted when $f > 0$ when mean is used to aggregate the gradients.

The failures can occur due to component or software failures and their probability increases with the scale of the system Wang et al. (2017); Tiwari et al. (2015); Nie et al. (2016). Reliability theory is used to analyze such failures, see Chapter 9 in Ross (2014), but for large-scale training, the distribution of total system failures is not independent over workers, making the total noise in gradients dependent and a key challenge for large-scale training. Moreover, even if there are no issues with the infrastructure, our work is motivated by the prevalence of data augmentation, including hand-chosen augmentations. Since number of parameters $n$ is often greater than number of samples, data augmentation improves the generalization capabilities of large-scale models under technical conditions Yang et al. (2019b); Heinze-Deml and Meinshausen (2017); Motiian et al. (2017). In particular, Adversarial training is a common technique that finds samples that are close to training

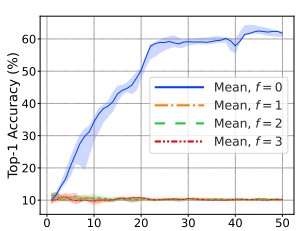

Figure 2: Tolerance to $f$ Byzantine workers for a non-robust aggregator (mean).

samples but classified as a different class at the current set of parameters, and then use such samples for parameter update purposes Addepalli et al. (2022). Unfortunately, computing adversarial samples is often difficult Wong et al. (2020), done using randomized algorithms Cohen et al. (2019) and so may introduce dependent (across samples) noise themselves. In other words, using adversarial training paradigm, or the so-called inner optimization can lead to noise in gradients, which can cause or simulate dependent "Byzantine" failures in the distributed context – since the final parameters are influenced by these samples.

**Available Computational Solutions.** Most existing open source implementations of $\mathcal{A}$ rely just on (functions of) pairwise distances to filter gradients from workers using suitable neighborhood based thresholding schemes, based on moment conditions Allouah et al. (2023a;b); Farhadkhani et al. (2022). While these may be a good strategy when the noise in samples/gradients is somewhat independent, these methods are suboptimal when the noise is dependent or nonlinear, especially when $n$ is large. Moreover, choosing discrete hyperparameters such as number of neighbors is impractical in our use cases since they hamper convergence of the overall training procedure. To mitigate the suboptimality of existing aggregation schemes, we explicitly estimate a subspace $Y$ spanned by "most" of the gradient workers, and then use this subspace to estimate that a **sparse** linear combination of $g_i$ gradients, acheiving robustness.

We present a new optimization based formulation for generalized gradient aggregation purposes in the context of distributed training of deep learning architectures, as shown in Figure 1.

**Summary of our Contributions.** From the theoretical perspective, we present a simple Maximum Likelihood Based estimation procedure for aggregation purposes, with novel regularization functions. Algorithmically, we argue that any procedure used to solve Flag Optimization can be directly used to obtain the optimal summary statistic $Y^*$ for our aggregation purposes. **Experimentally**, our results show resilience against Byzantine attacks, encompassing physical failures, while effectively managing the stochasticity arising from data augmentation schemes. In practice, we achieve a *significantly* ($\approx 20\%$) better accuracy on standard datasets. Our **implementation** offers substantial advantages in reducing communication complexity across diverse noise settings through the utilization of our novel aggregation function, making it applicable in numerous scenarios.

## 2 ROBUST AGGREGATORS AS ORTHOGONALITY CONSTRAINED OPTIMIZATION

In this section, we first provide the basic intuition of our proposed approach to using subspaces for aggregation purposes using linear algebra, along with connections of our approach standard eigendecomposition based denoising approaches. We then present our overall optimization formulation in two steps, and argue that it can be optimized using existing methods.

### 2.1 OPTIMAL SUBSPACE HYPOTHESIS FOR DISTRIBUTED DESCENT

We will use lowercase letters $y, g$ to denote vectors, and uppercase letters $Y, G$ to denote matrices. We will use **boldfont 1** to denote the vector of all ones in appropriate dimensions. Let $g_i \in \mathbb{R}^n$ be the gradient vector from worker $i$, and $Y \in \mathbb{R}^{n \times m}$ be an orthogonal matrix representation of a subspace that gradients could live in such that $m \leq p$.

Now, we may interpret each column of $Y$ as a basis function that act on $g_i \in \mathbb{R}^n$, i.e., $j-$th coordinate of $(Y^T g)_j$ for $1 \leq j \leq m$ is the application of $j-$th basis or column of $Y$ on $g$. Recall that by definition of dot product, we have that if $Y_{:,j} \perp g$, then $(Y^T g)_j$ will be close to zero. Equivalently, if $g \in \text{span}(Y)$, then $(Y^T g)^T Y^T g$ will be bounded away from zero, see Chapter 2 in Absil (2008). Assuming that $G \in \mathbb{R}^{n \times p}$ is the gradient matrix of $p$ workers, $YY^T G \in \mathbb{R}^{n \times p}$ is the reconstruction of $G$ using $Y$ as basis. That is, $i^{th}$ column of $Y^T G$ specifies the amount of gradient from worker $i$ as a function of $Y$, and high $l_2$ norm of $Y^T g_i$ implies that there is a basis in $Y$ such that $Y \not\perp g_i$. So it is easy to see that the average over columns of $YY^T G$ would give the final gradient for update. Please refer to Section C in the supplement for more detail.

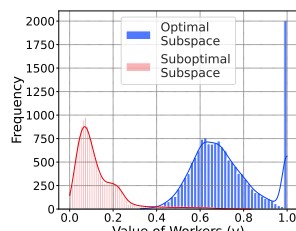

Figure 3: Distributions of Explained Variances on Minibatches

**Explained Variance of worker** $i$**.** If we denote $z_i = Y^T g_i \in \mathbb{R}^m$ representing the transformation of gradient $g_i$ to $z_i$ using $Y$, then, $0 \leq \|z_i\|_2^2 = z_i^T z_i = (Y^T g_i)^T Y^T g_i = g_i^T Y Y^T g_i$ is a scalar, and so is equal to its trace $\text{tr}\left(g_i^T Y Y^T g_i\right)$. Moreover, when $Y$ is orthogonal, we have $0 \leq \|z_i\|_2 = \|Y^T g_i\|_2 \leq \|Y\|_2 \|g_i\|_2 \leq \|g_i\|_2$ since the operator norm (or largest singular value) $\|Y\|_2$ of $Y$ is at most 1. Our main idea is to use $\|z_i\|_2^2, \|g_i\|_2^2$ to define the quality of the subspace $Y$ for aggregation, as is done in some previous works for Robust Principal Component Estimation Wright et al. (2009) –

the quantity $\|z_i\|_2^2/\|g_i\|_2^2$ is called as *Explained/Expressed* variance of subspace $Y$ wrt $i-$th worker Hein and Bühler (2010); Chakraborty et al. (2017) – we refer to $\|z_i\|_2^2/\|g_i\|_2^2$ as the "value" of $i-$th worker. In Figure 3, we can see from the spike near 1.0 that if we choose the subspace carefully (blue) as opposed to merely choosing the mean gradient (with unit norm) of all workers, then we can increase the value of workers.

**Advantages of Subspace based Aggregation.** We can see that using subspace $Y$, we can easily: 1. handle different number of gradients from each worker, 2. compute gradient reconstruction $YY^TG$ efficiently whenever $Y$ is constrained to be orthogonal $Y = \sum_i y_i y_i^T$ where $y_i$ is the $i-$th column of $Y$, otherwise have to use eigendecomposition of $Y$ to measure explained variance which can be time consuming. Please refer to the supplement for more details on reconstruction and why we require orthogonality constraint. In (practical) distributed settings, the quality (or noise level) of gradients in each worker may be different, **and/or** each worker may use a different batch size. In such cases, handcrafted aggregation schemes may be difficult to maintain, and fine-tune. For these purposes with an Orthogonal Subspace $Y$, we can simply reweigh gradients of worker $i$ according to its noise level, **and/or** use $g_i \in \mathbb{R}^{n \times b_i}$ where $b_i$ is the batch size of $i-$th worker with $\text{tr}(z_i^T z_i)$ instead.

**Why is optimizing over subspaces called "Flag" Optimization?** Recent optimization results suggest that we can exploit the finer structure available in Flag Manifold to specify $Y$ more precisely Monk (1959). For example, $Y \in \mathbb{R}^{n \times m}$ can be parametrized directly as a subspace of dimension $m$ or as a nested sequence of $Y_k \in \mathbb{R}^{n \times m_k}, k = 1, ..., K$ where $m_k < m_{k+1} \leq p \leq n$ such that $\text{span}(Y_k) \subseteq \text{span}(Y_{k+1})$ with $Y_K \in \mathbb{R}^{n \times m}$. When $m_{k+1} = m_k = 1$, we have the usual (real) Grassmanian Manifold (quotient of orthogonal group) whose coordinates can be used for optimization, please see Section 5 in Ye et al. (2022) for details. In fact, Mankovich et al. (2022) used this idea to extend median in one-dimensional vector spaces to different finite dimensional *subspaces* using the so-called chordal distance between them. In our distributed training context, we use the explained variance of each worker instead. Here, workers may specify dimensions along which gradient information is relevant for faster convergence – an advantage currently not available in existing aggregation implementations – which may be used for smart initialization also. *We use "Flag" to emphasize this additional nested structure available in our formulation for distributed training purposes.*

## 2.2 Approximate Maximum Likelihood Estimation of Optimal Subspace

Now that we can evaluate a subspace $Y$ on individual gradients $g_i$, we now show that finding subspace $Y$ can be formulated using standard maximum likelihood estimation principles Murphy (2022). Our formulation reveals that regularization is critical for aggregation especially in distributed training. In order to write down the objective function for finding optimal $Y$, we proceed in the following two steps:

**Step 1.** Assume that each worker provides a single gradient for simplicity. Now, denoting the value of information $v$ of worker $i$ by $v_i = \frac{z_i^T z_i}{g_i^T g_i}$, we have $v_i \in [0, 1]$. Now by assuming that $v_i$'s are observed from Beta distribution with $\alpha = 1$ and $\beta = \frac{1}{2}$ (for simplicity), we can see that the likelihood $\mathbb{P}(v_i)$ is,

$$\mathbb{P}(v_i) := \frac{(1 - v_i)^{-\frac{1}{2}}}{B(1, \frac{1}{2})} = \frac{\left(1 - \frac{z_i^T z_i}{g_i^T g_i}\right)^{-\frac{1}{2}}}{B(1, \frac{1}{2})}, \tag{2}$$

where $B(a, b)$ is the normalization constant. Then, the total log-likelihood of observing gradients $g_i$ as a function of $Y$ (or $v_i$'s) is given by taking the log of product of $\mathbb{P}(v_i)$'s as (ignoring constants),

$$\log\left(\prod_{i=1}^p \mathbb{P}(v_i)\right) = \sum_{i=1}^p \log\left(\mathbb{P}(v_i)\right) = -\frac{1}{2}\sum_{i=1}^p \log(1 - v_i). \tag{3}$$

**Step 2.** Now we use Taylor's series with constant $a > 0$ to approximate individual worker log-likelihoods $\log(1 - v_i) \approx a(1 - v_i)^{\frac{1}{a}} - a$ as follows: first, we know that $\exp\left(\frac{\log(v_i)}{a}\right) = v_i^{\frac{1}{a}}$. On the other hand, using Taylor expansion of $\exp$ about the origin (so large $a > 1$ is better), we have that $\exp\left(\frac{\log(v_i)}{a}\right) \approx 1 + \frac{\log(v_i)}{a}$. Whence, we have that $1 + \frac{\log(v_i)}{a} \approx v_i^{\frac{1}{a}}$ which immediately implies

that $\log(v_i) \approx a v_i^{\frac{1}{a}} - a$. So, by substituting the Taylor series approximation of $\log$ in Equation 3, we obtain the *negative* log-likelihood approximation to be *minimized* for robust aggregation purposes as,

$$-\log\left(\prod_{i=1}^{p}\mathbb{P}(v_i)\right) \approx \frac{1}{2}\sum_{i=1}^{p}\left(a\left(1-v_i\right)^{\frac{1}{a}} - a\right), \tag{4}$$

where $a > 1$ is a sufficiently large constant. In the above mentioned steps, the first step is standard. Our key insight is using Taylor expansion in (4) with a sufficiently large $a$ to eliminate $\log$ optimization which are known to be computationally expensive to solve, and instead solve *smooth $\ell_a, a > 1$* norm based optimization problems which can be done efficiently by modifying existing procedures Fornasier et al. (2011).

**Extension to general beta distributions, and gradients** $\alpha > 0, \beta > 0, g_i \in \mathbb{R}^{n \times k}$. Note that our derivation in the above two steps can be extended to any beta shape parameters $\alpha > 0, \beta > 0$ – there will be two terms in the final negative log-likelihood expression in our formulation (4), one for each $\alpha, \beta$. Similarly, by simply using $v_i = \text{tr}\left(g_i^T Y Y^T g_i\right)$ to define value of worker $i$ in equation (2), and then in our estimator in (4), we can easily handle multiple $k$ gradients from a single worker $i$ for $Y$.

## 2.3 Flag Aggregator for Distributed Optimization

It is now easy to see that by choosing $a = 2$, in equation (4), we obtain the negative loglikelihood (ignoring constants) as $\left(\sum_{i=1}^{p}\sqrt{1 - g_i^T Y Y^T g_i}\right)$ showing that Flag Median can indeed be seen as an Maximum Likelihood Estimator (MLE). In particular, Flag Median Mankovich et al. (2022) can be seen as an MLE of Beta Distribution with parameters $\alpha = 1$ and $\beta = \frac{1}{2}$. Recent results suggest that in many cases, MLE is ill-posed, and regularization is necessary, even when the likelihood distribution is Gaussian Karvonen and Oates (2023). So, based on the Flag Median estimator for subspaces, we propose an optimization based subspace estimator $Y^*$ for aggregation purposes. We formulate our Flag Aggregator (FA) objective function with respect to $Y$ as a *regularized* sum of likelihood based (or data) terms in (4) using trace operators $\text{tr}(\cdot)$ as the solution to the following constrained optimization problem:

$$\min_{Y:Y^T Y = I} A(Y) := \sum_{i=1}^{p}\sqrt{\left(1 - \frac{\text{tr}\left(Y^T g_i g_i^T Y\right)}{\|g_i\|_2^2}\right)} + \lambda \mathcal{R}(Y) \tag{5}$$

where $\lambda > 0$ is a regularization hyperparameter. In our analysis, and implementation, we provide support for two possible choices for $\mathcal{R}(Y)$:

(1) **Mathematical norms:** $\mathcal{R}(Y)$ can be a form of norm-based regularization other than $\|Y\|_{\text{Fro}}^2$ since it is constant over the feasible set in (5). For example, it could be convex norm with efficient convex approximation using trace functions such as, i.e. element-wise: $\sum_{i=1}^{n}\sum_{j=1}^{m}|Y_{ij}| \approx \text{tr}(Y^T Y)/\delta$ where $\delta > 0$ is the tolerance parameter for regularization subsumed within $\lambda$,

(2) **Data-dependent norms:** Following our subspace construction in Section 2.1, we may choose $\mathcal{R}(Y) = \frac{1}{p-1}\sum_{i,j=1,i\neq j}^{p}\sqrt{\left(1 - \frac{\text{tr}(Y^T(g_i-g_j)(g_i-g_j)^T Y)}{D_{ij}^2}\right)}$ where $D_{ij}^2 = \|g_i - g_j\|_2^2$ denotes the distance between gradient vectors $g_i, g_j$ from workers $i, j$. Intuitively, the pairwise terms in our loss function (5) favors subspace $Y$ that also reconstructs the pairwise vectors $g_i - g_j$ that are close to each other. So, by setting $\lambda = \Theta(p)$, that is, the pairwise terms dominate the objective function in (5). Hence, $\lambda$ regularizes optimal solutions $Y^*$ of (5) to contain $g_i$'s with low pairwise distance in its span – similar in spirit to AggregaThor in Damaskinos et al. (2019).

**Convergence of Flag Aggregator (FA) Algorithm 1.** With these, we can state our main algorithmic result showing that our FA (5) can be solved efficiently using standard convex optimization proof techniques. In particular, in supplement, we present a smooth Semi-Definite Programming (SDP) relaxation of FA in equation (5) using the Flag structure. We use the SDP relaxation of the MLE in (4) to argue that solving FA problem may be tractable since SDPs can be solved efficiently from the theoretical standpoint. With this, we can view the IRLS procedure in 1 as solving the low rank parametrization of the smooth SDP relaxation, thus guaranteeing fast convergence to second order optimal (local) solutions. Importantly, our SDP based proof works for any degree of approximation of the constant $a$ in equation (4) and only relies on smoothness of the loss function wrt $Y$, although speed of convergence is reduced for higher values of $a \neq 2$, see Chen et al. (2020a).

---

**Algorithm 1** Distributed SGD with proposed Flag Aggregator (FA) with elementwise $\ell_1$ regularization

---

**Input:** Number of workers $p$, loss functions $l_1, l_2, ..., l_p$, per-worker minibatch size $B$, learning rate schedule $\alpha_t$, initial parameters $w_0$, number of iterations T

**Output:** Updated parameters $w_T$ from any worker

1 **for** $t = 1$ *to* $T$ **do**
2     **for** $\mathfrak{p} = 1$ *to* $p$ *in parallel on machine* $\mathfrak{p}$ **do**
3        **Select a minibatch:** $i_{\mathfrak{p},1,t}, i_{\mathfrak{p},2,t},\ldots,i_{\mathfrak{p},B,t}$    $g_{\mathfrak{p},t} \leftarrow \frac{1}{B}\sum_{b=1}^{B} \nabla l_{i_{\mathfrak{p},b,t}}(w_{t-1})$
4     $G_t \leftarrow \{g_{1,t}, \cdots, g_{p,t}\}$ `// Parameter Server receives gradients from p workers`
5     $\hat{Y}_t \leftarrow \text{SVD}_m(G_t)$ with $\hat{G}_t$ is the Cholesky factor of $G_t D_t G^T + \lambda I$ where $I \in \mathbb{R}^{n \times n}$ identity matrix `// Do IRLS at the Parameter Server for` $\hat{Y}$
6     **Obtain gradient direction** $d_t$: $d_t = \frac{1}{p}\hat{Y}_t\hat{Y}_t^T G_t \mathbf{1}$ `// Compute, Send` $d_t$ `to all p machines`
7     **for** $\mathfrak{p} = 1$ *to* $p$ *in parallel on machine* $\mathfrak{p}$ **do**
8        **update model:** $w_t \leftarrow w_{t-1} - \alpha_t \cdot d_t$

9 **Return** $w_T$

---

**IRLS procedure in Algorithm 1.** IRLS is a standard optimization technique in which we substitute general norm functions with weighted euclidean norm functions. The key advantage of this substitution is that we may obtain closed form solution to the substituted euclidean norm version. Starting from a (random) feasible point $Y_{\text{old}}$, the weights are calculated with the general norm functions. Then, the solution $Y_{\text{new}}$ to the weighted euclidean norm optimization is obtained. This corresponds to one iteration in IRLS and repeating the above step with this new $Y_{\text{new}}$ corresponds to the IRLS procedure. For aggregation purposes in FA, in each iteration the square root function or more generally, the $a-$th root function in equation (4) is replaced by reweighted quadratic function which has a closed-form solution given by SVD.

**How is FA aggregator different from (Bulyan and Multi-Krum)?** Bulyan is a strong Byzantine resilient gradient aggregation rule for $p \geq 4f + 3$ where $p$ is the total number of workers and $f$ is the number of Byzantine workers. Bulyan is a two-stage algorithm. In the first stage, a gradient aggregation rule $R$ like coordinate-wise median Yin et al. (2018) or Krum Blanchard et al. (2017) is recursively used to select $\theta = p - 2f$ gradients. The process uses $R$ to select gradient vector $g_i$ which is closest to $R$'s output (e.g. for Krum, this would be the gradient with the top score, and hence the exact output of $R$). The chosen gradient is removed from the received set and added to the selection set $S$ repeatedly until $|S| = \theta$. The second stage produces the resulting gradient. If $\beta = \theta - 2f$, each coordinate would be the average of $\beta$-nearest to the median coordinate of the $\theta$ gradients in $S$. In matrix terms, if we consider $S \in \mathbb{R}^{p \times m}$ as a matrix with each column having one non-zero entry summing to 1, Bulyan would return $\frac{1}{m}\text{ReLU}(GS)\mathbf{1}_m$, where $\mathbf{1}_m \in \mathbb{R}^m$ is the vector of all ones, while FA would return $\frac{1}{p}YY^T G\mathbf{1}_p$. Importantly, the gradient matrix is being right-multiplied in Bulyan, but left-multiplied in FA, before getting averaged. While this may seem like a discrepancy, in supplement we show that by observing the optimality conditions of (5) wrt $Y$, we show that $\frac{1}{m}YY^T G$ can be seen as a right multiplication by a matrix parametrized by lagrangian multipliers associated with the orthogonality constraints in (5). This means it should be possible to combine both approaches for faster aggregation.

## 3 EXPERIMENTS

In this section, we conduct experiments to test our proposed FA in the context of distributed training in two testbeds. First, to test the performance of our FA scheme solved using IRLS (Flag Mean) on standard Byzantine benchmarks. Then, to evaluate the ability of existing state-of-the-art gradient aggregators we augment data via two techniques that can be implemented with Sci-kit package.

**Implementation Details.** We implement FA in Pytorch Paszke et al. (2019), which is popular but does not support Byzantine resilience natively. We adopt the parameter server architecture and employ Pytorch's distributed RPC framework with TensorPipe backend for machine-to-machine communication. We extend Garfield's Pytorch library Guerraoui et al. (2021) with FA and limit our

IRLS convergence criteria to a small error, $10^{-10}$, or 5 iterations of flag mean for SVD calculation. We set $m = \lceil \frac{p+1}{2} \rceil$.

## 3.1 SETUP

**Baselines:** We compare FA to several existing aggregation rules: (1) coordinate-wise **Trimmed Mean** Yin et al. (2018) (2) coordinate-wise **Median** Yin et al. (2018) (3) mean-around-median **(MeaMed)** Xie et al. (2018a) (4) **Phocas** Xie et al. (2018b) (5) **Multi-Krum** Blanchard et al. (2017) (6) **Bulyan** El Mhamdi et al. (2018). In the supplement, we also compare FA to using the (7) **top-$m$ principal components** of the gradient matrix, (8) **RESAM** Farhadkhani et al. (2022), and (9) **CGE** Gupta et al. (2021) as baselines.

**Accuracy:** The fraction of correct predictions among all predictions, using the test dataset (top-1 cross-accuracy).

**Testbed:** We used 4 servers as our experimental platform. Each server has 2 Intel(R) Xeon(R) Gold 6240 18-core CPU @ 2.60GHz with Hyper-Threading and 384GB of RAM. Servers have a Tesla V100 PCIe 32GB GPU and employ a Mellanox ConnectX-5 100Gbps NIC to connect to a switch. We use one of the servers as the parameter server and instantiate 15 workers on other servers, each hosting 5 worker nodes, unless specified differently in specific experiments. For the experiments designed to show scalability, we instantiate 60 workers.

**Dataset and model:** We focus on the image classification task since it is a widely used task for benchmarking in distributed training Chilimbi et al. (2014). We train ResNet-18 He et al. (2016) on CIFAR-10 Krizhevsky (2009) which has 60,000 $32 \times 32$ color images in 10 classes. For the scalability experiment, we train a CNN with two convolutional layers followed by two fully connected layers on MNIST LeCun and Cortes (2010) which has 70,000 $28 \times 28$ grayscale images in 10 classes. We also run another set of experiments on Tiny ImageNet Le and Yang (2015) in the supplement. We use SGD as the optimizer, and cross-entropy to measure loss. The batch size for each worker is 128 unless otherwise stated. Also, we use a learning decay strategy where we decrease the learning rate by a factor of 0.2 every 10 epochs.

**Threat models:** We assess the performance of FA in the presence of various categories of Byzantine workers. These categories encompass scenarios where workers either transmit gradients chosen uniformly at random in $[0, 1)$ with $\mu = 0.5$ and $\sigma = \sqrt{\frac{1}{12}}$ or provide incomplete tensors. These scenarios serve as representations of errors in the physical settings. Additionally, workers may employ non-linear augmented data, as described below. In the supplement, we also evaluate FA when Byzantine workers send 10x amplified sign-flipped gradients Allen-Zhu et al. (2021) or send the gradients based on the Fall of Empires attack with $\epsilon = 0.1$ Xie et al. (2020).

**Evaluating resilience against nonlinear data augmentation:** In order to induce Byzantine behavior in our workers we utilize ODE solvers to approximately solve 2 non-linear processes, Lotka Volterra Kelly (2016) and Arnold's Cat Map Bao and Yang (2012), as augmentation methods. Since the augmented samples are deterministic, albeit nonlinear functions of training samples, the "noise" is dependent across samples.

In **Lotka Volterra**, we use the following linear gradient transformation of 2D pixels:

$$(x, y) \rightarrow (\alpha x - \beta xy, \delta xy - \gamma y),$$

where $\alpha, \beta, \gamma$ and $\delta$ are hyperparameters. We choose them to be $\frac{2}{3}, \frac{4}{3}, -1$ and $-1$ respectively.

Second, we use a *nonsmooth* transformation called **Arnold's Cat Map** as a data augmentation scheme. Once again, the map can be specified using a two-dimensional matrix as,

$$(x, y) \rightarrow \left( \frac{2x + y}{N}, \frac{x + y}{N} \right) \mod 1,$$

where `mod` represents the modulus operation, $x$ and $y$ are the coordinates or pixels of images and $N$ is the height/width of images (assumed to be square). We also used a smooth approximation of the Cat Map obtained by approximating the `mod` function as,

$$(x, y) \rightarrow \frac{1}{n} \left( \frac{2x + y}{(1 + \exp(-m \log(\alpha_1))}, \frac{x + y}{(1 + \exp(-m \log(\alpha_2))} \right),$$

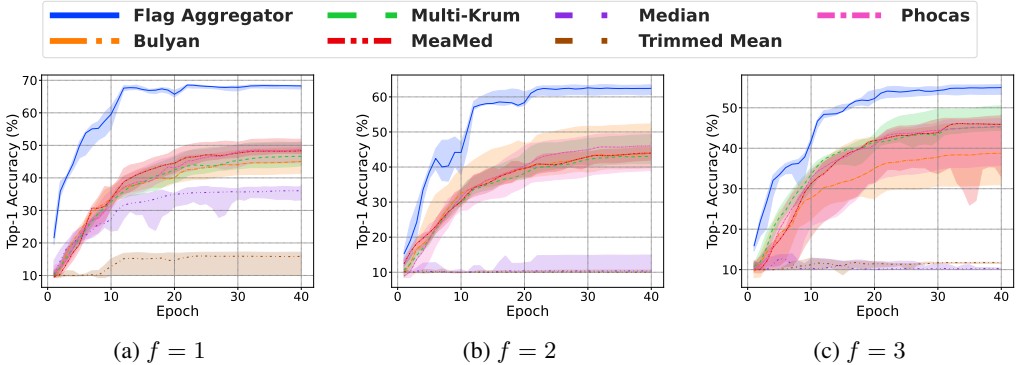

Figure 4: Tolerance to the number of Byzantine workers for robust aggregators for batch size 128.

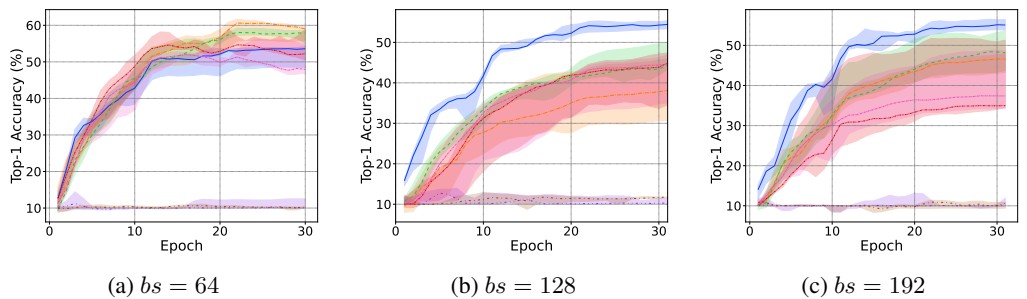

Figure 5: Marginal utility of larger batch sizes under a fixed noise level $f = 3$.

where $\alpha_1 = \frac{2x+y}{n}$, $\alpha_2 = \frac{x+y}{n}$, and $m$ is the degree of approximation, which we choose to be 0.95 in our data augmentation experiments. Please refer to the appendix F.3 where we explain our implementation for nonlinear data augmentation routines.

## 3.2 RESULTS

**Tolerance to the number of Byzantine workers:** In this experiment, we show the effect of Byzantine behavior on the convergence of different gradient aggregation rules in comparison to FA. Byzantine workers send uniformly random gradients in $[0, 1]$ with coordinatewise mean $0.5$ and standard deviation $\sqrt{\frac{1}{12}}$ and we vary the number of these workers from 1 to 3. Figure 4 shows that for some rules, i.e. Trimmed Mean, the presence of even a single Byzantine worker has a catastrophic impact. For other rules, as the number of Byzantine workers increases, filtering out the outliers becomes more challenging because the amount of noise increases. Regardless, FA remains more robust

**Marginal utility of larger batch sizes under a fixed noise level:** We empirically verified the batch size required to identify our optimal $Y^*$ - the FA matrix at each iteration. In particular, we fixed the noise level to $f = 3$ Byzantine workers and varied batch sizes. Similar to the previous experiment, Byzantine workers send uniformly random gradients. We show the results in Figure 5. **Our results indicate that, in cases where a larger batch size is a training requirement, FA achieves a significantly better accuracy compared to the existing state of the art aggregators.** This may be useful in some large scale vision applications, see Keskar et al. (2017); You et al. (2019) for more details. Empirically, we can already see that our spectral relaxation to identify gradient subspace is effective in practice in all our experiments.

**Tolerance to communication loss:** To analyze the effect of unreliable communication channels between the workers and the parameter server on convergence, we design an experiment where the physical link between some of the workers and the parameter server randomly drops a percentage of packets. Here, we set the loss rate of three links to 10% i.e., there are 3 Byzantine workers in our setting. The loss is introduced using the *netem* queuing discipline in Linux designed to emulate the properties of wide area networks Hsieh et al. (2017). The two main takeaways in Figure 6a are:

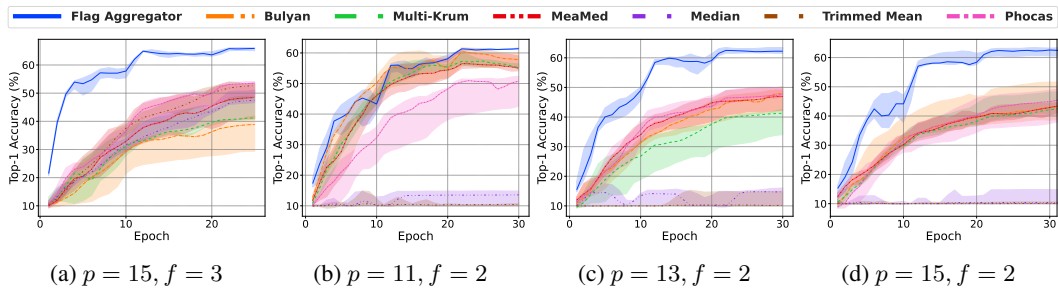

(a) $p = 15, f = 3$     (b) $p = 11, f = 2$     (c) $p = 13, f = 2$     (d) $p = 15, f = 2$

Figure 6: We present results under two different gradient attacks. The attack in (a) corresponds to dropping $10\%$ of gradients from $f$ workers. The attacks in (b)-(d) correspond to generic $f$ workers sending random gradient vectors, i.e. we fix the noise level while adding more workers.

> 1. FA converges to a significantly higher accuracy than other aggregators, and thus is more robust to unreliable underlying network transports.
>
> 2. Considering time-to-accuracy for comparison, FA reaches a similar accuracy in less total number of training iterations, and thus is more robust to slow underlying network transports.

**Analyzing the marginal utility of additional workers.** To see the effect of adding more workers to a fixed number of Byzantine workers, we ran experiments where we fixed $f$, and increased $p$. Our experimental results shown in Figures 6b-6d indicate that our FA algorithm possesses strong resilience property for reasonable choices of $p$.

**The effect of having augmented data during training in Byzantine workers:** Figure 7 shows FA can handle nonlinear data augmentation in a much more stable fashion. Please see supplement for details on the level of noise, and exact solver settings that were used to obtain augmented images.

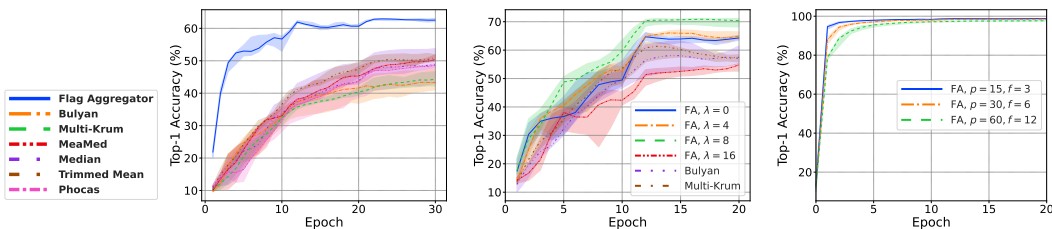

Figure 7: Accuracy of using augmented data in $f = 3$ workers

Figure 8: CIFAR-10 with ResNet-18, $p = 7$, and $f = 1$

Figure 9: Scaling FA to larger setups

**The effect of the regularization parameter in FA:** The data-dependent regularization parameter $\lambda$ in FA provides flexibility in the loss function to cover aggregators that benefit from pairwise distances such as Bulyan and Multi-Krum. To verify whether varying $\lambda$ can interpolate Bulyan and Multi-Krum, we change $\lambda$ in Figure 8. We can see when FA improves or performs similarly for a range of $\lambda$. Here, we set $p$ and $f$ to satisfy the strong Byzantine resilience condition of Bulyan, i.e, $p \geq 4f + 3$.

**Scaling out to real-world situations with more workers:** In distributed ML, $p$ and $f$ are usually large. To test high-dimensional settings commonly dealt in Semantic Vision with our FA, we used ResNet-18. Now, to specifically test the scalability of FA, we fully utilized our available GPU servers and set up to $p = 60$ workers (up to $f = 14$ Byzantine) with the MNIST dataset and a simple CNN with two convolutional layers followed by two fully connected layers (useful for simple detection). Figure 9 shows evidence that FA is feasible for larger setups.

## 4 CONCLUSION

In this paper we proposed Flag Aggregator (FA) that can be used for robust aggregation of gradients in distributed training. FA is an optimization-based subspace estimator that formulates aggregation as a Maximum Likelihood Estimation procedure using Beta densities. We perform extensive evaluations of FA and show it can be effectively used in providing Byzantine resilience for gradient aggregation. Using techniques from convex optimization, we theoretically analyze FA and with tractable relaxations show its amenability to be solved by off-the-shelf solvers or first-order reweighing methods.

## 5 ACKNOWLEDGEMENTS

We are grateful to UIC-ICR start-up funds and Discovery Partners Institute for supporting our work. We thank anonymous ICLR 2024 reviewers for their time, suggestions, and constructive criticism throughout the review process which ultimately led to an improved version of our submission.

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

# A BACKGROUND AND RELATED WORK

Researchers have approached the Byzantine resilience problem from two main directions. In the first class of works, techniques such as geometric median and majority voting try to perform robust aggregation Damaskinos et al. (2019), Bernstein et al. (2019), Blanchard et al. (2017). The other class of works uses redundancy and assigns each worker redundant gradient computation tasks Rajput et al. (2019), Chen et al. (2018).

From another aspect, robustness can be provided on two levels. In weak Byzantine resilience methods such as Coordinate-wise median Yin et al. (2018) and Krum Blanchard et al. (2017), the learning is guaranteed to converge. In strong Byzantine resilience, the learning converges to a state as the system would converge in case no Byzantine worker existed. Draco Chen et al. (2018) and Bulyan El Mhamdi et al. (2018) are examples of this class. Convergence analysis of iterated reweighing type algorithms has been done for specific problem classes. For example, Kümmerle et al. (2021); Adil et al. (2019) show that when IRLS is applied for sparse regression tasks, the iterates can converge linearly. Convergence analysis of matrix factorization problems using IRLS-type schemes has been proposed before, see Fornasier et al. (2011); Chen et al. (2020a).

It is well known that data augmentation techniques help in improving the generalization capabilities of models by adding more identically distributed samples to the data pool. Yang et al. (2019b); Heinze-Deml and Meinshausen (2017); Motiian et al. (2017). The techniques have evolved along with the development of the models, progressing from the basic ones like rotation, translation, cropping, flipping, injecting Gaussian noise Bishop (1995) etc., to now the sophisticated ones (random erasing/masking Zhong et al. (2020), cutout DeVries and Taylor (2017) etc.). Multi-modal learning setups Goyal et al. (2017); Chen et al. (2020b); Huang et al. (2021), use different ways to combine data of different modalities (text, images, audio etc.) to train deep learning networks.

# B TRACTABILITY OF COMPUTING FLAG AGGREGATORS

In this section, we characterize the computational complexity of solving the Flag Aggregation problem via IRLS type schemes using results from convex optimization. First, we present a tight convex relaxation of the Flag Median problem by considering it as an instantiation of rank constrained optimization problem. We then show that we can represent our convex relaxation as a Second Order Cone Program which can be solved using off-the-shelf solvers Lobo et al. (1998). Second, we argue that approximately solving such rank constrained problems in the factored space is an effective strategy using new results from Bhojanapalli et al. (2018) which builds on asymptotic convergence in Fornasier et al. (2011). Our results highlight that the Flag Median problem can be approximately solved using smooth optimization techniques, thus explaining the practical success of an IRLS type iterative solver.

**Interpreting Flag Aggregator (equation 5 in the main paper) in the Case** $m = 1$**.** We first present a convex reformulation of the Flag Aggregator problem (5) in the case when the number of subspaces (or columns) is equal to 1. To make the exposition easier, we will also assume that $\lambda = 0$. With these assumptions, and using the fact that $\|y\|_2 = 1$, each term in the objective function of our FA aggregator in (5) can be rewritten as,

$$\sqrt{(1 - (y^T \tilde{g}_i))^2)} = \sqrt{y^T \left(I - \tilde{g}_i \tilde{g}_i^T\right) y} = \|\tilde{B}_i y\|_2, \qquad (6)$$

where we use the notation $\tilde{g}_i = g_i / \|g_i\|$ to denote the normalized worker gradients, $I \in \mathbb{R}^{n \times n}$ is the $n \times n$ identity matrix, and $\tilde{B}_i$ is the square root of the matrix $I - \tilde{g}_i \tilde{g}_i^T$. Observe that we can rewrite all the terms in equation (5) in the main paper in a similar fashion as in (6). Furthermore, by relaxing the feasible set to the $n-$Ball given by $\{y \in \mathbb{R}^n : \|y\|_2 \leq 1\}$, we obtain a Second Order Cone Programming (SOCP) relaxation of our FA problem in (5). SOCP problems can be solved using off-the-shelf packages with open source optimization solvers for gradient aggregation purposes in small scale settings, that is, when the number of parameters $n \approx 10^4$ Shen et al. (2021); Kleiner et al. (2010); Mittelmann (2003). Our convex reformulation immediately yields insights on why reweighing type algorithm that was proposed in Mankovich et al. (2022) works well in practice – for example see Section 3 in Vanderbei and Yurttan (1998) in which various smoothing functions similar to the Flag Median (square) based smoothing are listed as options. More generally, our SOCP

relaxation shows that if the smoothed version can be solved in closed form (or efficiently), then a reweighing based algorithm can be safely considered a viable candidate for aggregation purposes.

**Tractable Reformulations when $m > 1$ for Aggregation Purposes.** Note that for any feasible $Y$ such that $Y^T Y = I$, we have that the $\text{tr}(Y) = m$.

*Remark* B.1 (Parametrizing Subspaces using $Y$.). This assumption is without loss of generality. To see this, first note that in general, a (nondegenerate) subspace $\mathcal{S}$ of a vector space $\mathcal{V}$ is defined as a subset of $\mathcal{V}$ that is closed under linear combinations. Fortunately, in finite dimensions, we can represent $\mathcal{S}$ as a rectangular matrix $M$ by Fundamental theorem of linear algebra. So, we simply use $Y$ to represent the basis of this matrix $M$ that represents the subspace $\mathcal{S}$ in our FA formulation.

Using this, we can rewrite each term in the objective function of our FA aggregator in equation (5) in the main paper as,

$$\sqrt{\text{tr}\left(Y^T \left(\frac{I}{m} - \frac{g_i g_i^T}{\|g_i\|_2^2}\right) Y\right)} = \sqrt{\text{tr}\left(Y^T M_i Y\right)}, \tag{7}$$

where $M_i = M_i^T, i = 1, ..., p$ is symmetric matrix with *at most* one negative eigenvalue. Optimization problems involving quadratic functions with negative eigenvalues can be solved globally, in some cases Luo et al. (2019); Shapiro and Botha (1988). We consider methods that can efficiently (say in polynomial running time in $n$) provide solutions that are locally optimal. In order to do so, we consider the Semi Definite programming relaxation obtained by introducing matrix $Z \succeq 0 \in \mathbb{R}^{nm \times nm}$ to represent the term $YY^T$, constrained to be rank one, and such that $\text{tr}(Z) = m$.

By using $\mathbf{vec}(Y) \in \mathbb{R}^{nm}$ to denote the vector obtained by stacking columns of $Z$, when $m > 1$, we obtain a trace norm constrained SOCP. Importantly, objective function can be written as a sum of terms of the form,

$$\sqrt{\mathbf{vec}(Y)^T (I \otimes M_i) \mathbf{vec}(Y)} = \sqrt{\text{tr}\left(Z^T (I \otimes M_i)\right)}, \tag{8}$$

where $\otimes$ denotes the usual tensor (or Kronecker) product between matrices.

**Properties of lifted formulation in (8).** There are some advantages to the loss function as specified in our reformulation (8). First, note that then our relaxation coincides with the usual trace norm based Goemans-Williamson relaxation used for solving Max-Cut problem with approximation guarantees Goemans and Williamson (1995). Albeit, our objective function is not linear, and to our knowledge, it is not trivial to extend the results to nonlinear cases such as ours in (8). Moreover, even when $M_i \succeq 0$, the $\sqrt{\cdot}$ makes the relaxation *nonconvex*, so it is not possible to use off-the-shelf disciplined convex programming software packages such as CVXPy Diamond and Boyd (2016); Agrawal et al. (2018). Our key observation is that away from 0, $\sqrt{\cdot}$ is a *differentiable* function. Hence, the objective function in (8) is differentiable with respect to $Z$.

*Remark* B.2 (Using SDP relaxation for Aggregation.). In essence, if the optimal solution $Z^*$ to the SDP relaxation is a rank one matrix, then by rank factorization theorem, $Z^*$ can be written as $Z^* = \mathbf{vec}(Y^*)\mathbf{vec}(Y^*)^T$ where $\mathbf{vec}(Y^*) \in \mathbb{R}^{mn \times 1}$. So, after reshaping, we can obtain our optimal subspace estimate $Y^* \in \mathbb{R}^{m \times n}$ for aggregation purposes. In the case the optimal $Z^*$ is not low rank, we simply use the largest rank one component of $Z^*$, and reshape it to get $Y^*$.

## C The necessity of orthogonality constraint for efficiency

This necessity is a folklore result that can be found in various places, but we provide a formal proof here for completeness. Given the gradient matrix $G$ and subspace $Y$, projection of $G$ onto $Y$ is given by $\mathbf{P} = Y^T G$. The entry $\mathbf{P}_{ji}$ has the amount (measured using dot product) of $g_i$ along $y_j$. So, $YP$ gives us the reconstruction of $G$ using each column of $Y$. By reconstruction we mean that the matrix $YY^T G$ is the best (or optimal) $m-$rank reconstruction of $G$ – here optimality is with respect to Squared $\ell_2$ norm which is also known as Mean Reconstruction Error (MSE). In detail, we are given with gradient matrix $G$, and $y_j, j = 1, ..., m$ such that $y_j$'s are orthonormal, that is, $y_j^T y_{j'} = 1$ if $j = j'$, and 0 otherwise. Since each column of $G$ is multiplied by the Projection matrix $YY^T$ separately, we consider each $g_i$ individually.

(i) Case 1: $m = 1$, so we are given with just one $y$ such that $\|y\|_2 = 1$. Then projecting $g_i$ onto $y$ in MSE is the solution to a 1-d optimization problem:

$$\arg\min_{\mathbf{p} \in \mathbb{R}} \left[ MSE(\mathbf{p}) := \|g_i - \mathbf{p}y\|_2^2 = \|g_i\|^2 - 2\mathbf{p}g_i^T y + \mathbf{p}^2 \|y\|_2^2 \right] = \frac{g_i^T y}{\|y\|_2^2} = g_i^T y, \qquad (9)$$

where we used the fact that $\|y\|_2 = 1$ in the last line. So the reconstruction is given by scaling $y$ by the optimal $\mathbf{p} = g_i^T y$. It turns out that this calculation can be performed with each basis as we show in the next case.

(ii) Case 2: $m > 1$, so we are given $m$ pairwise orthonormal vectors and similar to previous case we have to determine the $m$ projection coefficients for each $g_i$. Given $g_i$, we determine $\mathbf{p} \in \mathbb{R}^m$ as follows:

$$\arg\min_{\mathbf{p}_1, \cdots, \mathbf{p}_m} \left[ MSE(\mathbf{p}_1, \cdots, \mathbf{p}_m) := \left\| g_i - \sum_{j=1}^m \mathbf{p}_j y_j \right\|_2^2 = \|g_i\|_2^2 - 2\sum_{j=1}^m \mathbf{p}_j g_i^T y_j + \sum_{j=1}^m \mathbf{p}_j^2 \|y_j\|_2^2 \right]$$
$$(10)$$

where we used orthogonality relationship in the last equality. By setting $\nabla_{\mathbf{p}_j}(MSE) = 0$ we see that the reconstruction problem decomposes to $m$ 1-d optimization problems each with closed form solutions $\mathbf{p}_j = \frac{g_i^T y_j}{\|y_j\|^2} = g_i^T y_j, j = 1, \ldots, m$ as in the previous case. So in this case, the reconstruction is given by $\sum_j \mathbf{p}_j y_j = \sum_j y_j y_j^T g_i = YY^T g_i$

This illustrates why we require orthogonality constraints since otherwise, reconstruction might be computationally expensive. Note that $Y^T Y = I$ does not imply $YY^T = I$ since $m < n$. In literature, the matrix $YY^T$ is often called as the family of Projection matrices (not the $Y^T G$ as we do here) since $(YY^T)^2 = YY^T YY^T = YIY^T = YY^T$ for any orthonormal $Y$.

# D    SOLVING FLAG AGGREGATION EFFICIENTLY

**Convergence Analysis when** $m = 1$**.** Note that for the case $m = 1$, that is, FA provides unit vector $y \in \mathbb{R}^n$ to get aggregated gradient as $yy^T G$, we can use smoothness based convergence results in nonconvex optimization, for example, please see Jain et al. (2017). We believe this addresses most of the standard training pipelines used in practice. Now, we focus on the case with $m > 1$.

Now that we have a smooth reformulation of the aggregation problem that we would like to solve, it is tempting to solve it using first order methods. However, naively applying first order methods can lead to slow convergence, especially since the number of decision variables is now increased to $m^2 n^2$. Standard projection oracles for trace norm require us to compute the full Singular Value Decomposition (SVD) of $Z$ which becomes computationally expensive even for small values of $m, n \approx 10$.

Fortunately, recent results show that the factored form smooth SDPs can be solved in polynomial time using gradient based methods. That is, by setting $Z = \mathbf{vec}(Y)\mathbf{vec}(Y)^T$, and minimizing the loss functions $L_i(Y) = \sqrt{\mathbf{vec}(Y)^T (I \otimes M_i) \mathbf{vec}(Y)}$ with respect to $Y$, we have that the set of locally optimal points coincide, see Bhojanapalli et al. (2018). Moreover, we have the following convergence result for first order methods like Gradient Descent that require SVD of $n \times p$ matrices:

**Lemma D.1.** *If $L_i$ are $\kappa_i-$smooth, with a $\eta_i-$lipschitz Hessian, then projected gradient descent with constant step size converges to a locally optimal solution to* (8) *in $\tilde{O}(\kappa/\epsilon^2)$ iterations where $0 \leq \epsilon \leq \kappa^2/\eta$ is a error tolerance parameter, $\kappa = \max_i \kappa_i$, and $\eta = \max_i \eta_i$.*

Above lemma D.1 says that gradient descent will output an aggregation $Y$ that satisfies second order sufficiency conditions with respect to smooth reformulated loss function in (8). All the terms inside $\tilde{O}$ in lemma D.1 are logarithmic in dimensions $m, n$, lipschitz constant $L$, and accuracy parameter $\epsilon$.

*Remark* D.2 (Numerical Considerations.). Note that the lipschitz constant $\kappa$ of the overall objective function depends on $M_i$. That is, when $M_i$ has negative eigenvalues, then $\kappa$ can be high due to the square root function. We can consider three related ways to avoid this issue. First, we can choose a value $m' > m$ in our trace constraint such that $M_i \succeq 0$. Similarly, we can expand (8) (in $\sqrt{\cdot}$) as outer

product of columns of $Y$ suggesting that $\tilde{g}\tilde{g}^T$ term need to be normalized by $m$, thus making $M_i \succeq 0$. Secondly, we can consider adding a quadratic term such as $\|Y\|_{\text{Fro}}^2$ to make the function quadratic. This has the effect of decreasing $\kappa$ and $\eta$ of the objective function for optimization. Finally, we can use $m_i = \max(k_i, m)$ instead of $\min$ in defining the loss function as in Mankovich et al. (2022) which would also make $M_i \succeq 0$.

## E    PROOF OF LEMMA D.1 WHEN $m > 1$.

We provide the missing details in Section D when $m > 1$. To that end, we will assume that each worker $i$ provides the server with a list of $k_i$ gradients, that is, $g_i \in \mathbb{R}^{n \times k}$ – a strict generalization of the case considered in the main paper (with $k = 1$), that may be useful independently. Note that in Mankovich et al. (2022), these $g_i$'s are assumed to be subspaces whereas we do not make that assumption in our FA algorithm.

Now, we will show that the RHS in equation (7) and LHS in equation (8) are equivalent. For that, we need to recall an elementary linear algebra fact relating tensor/Kronecker product, and tr operator. Recall the definition of Kronecker product:

**Definition E.1.** Let $A \in \mathbb{R}^{d_1 \times d_2}, B \in \mathbb{R}^{e_1 \times e_2}$, then $A \otimes B \in \mathbb{R}^{d_1 e_1 \times d_2 e_2}$ is given by,

$$A \otimes B := \begin{bmatrix} a_{1,1}B & \dots & a_{1,d_2}B \\ \vdots & \ddots & \vdots \\ a_{d_1,1}B & \dots & a_{d_1,d_2}B \end{bmatrix}, \tag{11}$$

where $a_{i,j}$ denotes the entry at the $i-$th row, $j-$th column of $A$.

**Lemma E.2** (Equivalence of Objective Functions.)**.** *Let $Y \in \mathbb{R}^{n \times m}$, $g \in \mathbb{R}^{n \times k}$ (so, $M \in \mathbb{R}^{n \times n}$). Then, we have that,*

$$tr\left(Y^T gg^T Y\right) := tr\left(Y^T M Y\right) = \mathbf{vec}(Y)^T \left(I \otimes M\right) \mathbf{vec}(Y), \tag{12}$$

*where $I \in \mathbb{R}^{m \times m}$ is the identity matrix.*

*Proof.* Using the definition of tensor product in equation (11), we can simplify the right hand side of equation (12) as,

$$\mathbf{vec}(Y)^T \left(I \otimes M\right) \mathbf{vec}(Y) = [y_{11}, \cdots y_{n1}, \cdots, y_{1m}, \cdots, y_{nm}] \begin{bmatrix} M & 0 & \dots & 0 \\ \vdots & M & \dots & \vdots \\ \vdots & \vdots & \ddots & \vdots \\ 0 & \cdots & \cdots & M \end{bmatrix} \begin{bmatrix} y_{11} \\ \vdots \\ y_{n1} \\ \vdots \\ y_{1m} \\ \vdots \\ y_{mn} \end{bmatrix}$$

$$= \sum_{j=1}^{m} y_j^T M y_j$$

$$= \sum_{j=1}^{m} \text{tr}\left(y_j y_j^T M\right) = \text{tr}\left(\sum_{j=1}^{m} y_j y_j^T M\right) = \text{tr}\left(\left(\sum_{j=1}^{m} y_j y_j^T\right) M\right) \tag{13}$$

$$= \text{tr}\left(YY^T M\right) = \text{tr}\left(Y^T M Y\right), \tag{14}$$

where we used the cyclic property of trace operator $\text{tr}(\cdot)$ in equations (13), and (14) that is, $\text{tr}(ABC) = \text{tr}(CAB) = \text{tr}(BCA)$ for any dimension compatible matrices $A, B, C$. $\square$

### E.1    PROOF OF LEMMA D.1

Recall that, given $\tilde{M}_i = I \otimes M_i$, the lifted cone programming relaxation of FA can be written as,

$$\min_Z \sum_i \sqrt{\text{tr}(Z^T \tilde{M}_i)} \quad \text{s.t.} \quad Z \succeq 0, \ \text{tr}(Z) = m, Z = Z^T, \tag{15}$$

where $m$ is the rank of $Z$ or number of columns of $Y$. We now use the above Lemma E.2 to show that the objective function with respect to $Z$ in the lifted formulation is smooth which gives us the desired convergence result in Lemma D.1.

*Proof.* let $\tilde{\kappa}_i > 0$,

$$\frac{\partial \sqrt{\text{tr}(Z^T \tilde{M}_i) + \tilde{\kappa}_i}}{\partial Z} = \frac{1}{2\sqrt{\tilde{\kappa}_i + \text{tr}(Z^T \tilde{M}_i)}} \tilde{M}_i, \tag{16}$$

where $\tilde{M}_i = I \otimes M_i$ as in equation (8). Now, since $\tilde{M}_i$ is constant with respect to $Z$, the gradient term is affected only through a scalar $\sqrt{\text{tr}(Z^T \tilde{M}_i) + \tilde{\kappa}_i}$. So the largest magnitude or entrywise $\ell_\infty$-norm of the Hessian is given by,

$$\left| \frac{\partial \frac{1}{\sqrt{\text{tr}(Z^T \tilde{M}_i) + \tilde{\kappa}_i}} \|\tilde{M}_i\|_\infty}{\partial Z} \right| = \frac{\|\tilde{M}_i\|_\infty}{2\sqrt{\left(\text{tr}(Z^T \tilde{M}_i) + \tilde{\kappa}_i\right)^3}}. \tag{17}$$

Now, we will argue that the gradient and hessian are lipschitz continuous in the lifted space. Since any feasible $Z \succeq 0$ is positive semidefinite, if $\tilde{M}_i \succeq 0$, then the scalar $\text{tr}(Z^T \tilde{M}_i)$ is at least $m \cdot \lambda_{mn}^{\tilde{M}_i}$ where $\lambda_{mn}^{\tilde{M}_i}$ is the smallest (or $mn-$th) eigenvalue of $\tilde{M}_i$. So, we can choose $\tilde{\kappa}_i = 0 \forall i$. If not, then there is a negative eigenvalue, possibly repeated. So, the gradient might not exist. In cases where $\tilde{M}_i$ has negative eigenvalues, we can choose $\tilde{\kappa}_i = \tilde{\kappa} = \left| \min_i \min \left( \lambda_{mn}^{\tilde{M}_i}, 0 \right) \right|$. With these choices, we have that the gradient of the objective function in (8) is lipschitz continuous. By a similar analysis using the third derivative, we can show that Hessian is also lipschitz continuous with respect to $Z$. In other words, all the lipschitz constant of both the gradient and hessian of our overall objective function is controlled by $\tilde{\kappa} > 0$. Hence, we have all conditions satisfied required for Lemma 1 in Bhojanapalli et al. (2018), and we have our convergence result for FA in the factored space of $\mathbf{vec}(Y)$. $\square$

Few remarks are in order with respect to our convergence result. First, **is the choice $\tilde{\kappa}$ important for convergence?** Our convergence result shows that a perturbed objective function $\text{tr}(Z^T \tilde{M}_i) + \tilde{\kappa}$ has the same second order stationary points as that of the objective function in the factored form formulated using $Y$ (or $\mathbf{vec}(Y)$). We can avoid this perturbation argument if we explicitly add constraints $\text{tr}(Z^T \tilde{M}_i) \geq 0$, since projections on linear constraints can be performed efficiently exactly (sometimes) or approximately. Note that these constraints are natural since it is not possible to evaluate the square root of a negative number. Alternatively, we can use a smooth approximate approximation of the absolute values $\sqrt{\left| \text{tr}(Z^T \tilde{M}_i) \right|}$. In this case, it is easy to see from (16), and (17) that the constants governing the lipschitz continuity as dependent on the absolute values of the minimum eigenvalues, as expected. In essence, no, the choice of $\tilde{\kappa}$ does not affect the nature of landscape – approximate locally optimal points remain approximately locally optimal. In practice, we expect the choice of $\tilde{\kappa}$ to affect the performance of first order methods.

Second, **can we assume $\tilde{M}_i \succ 0$ for gradient aggregation purposes?** Yes, this is because, when using first order methods to obtain locally optimal solution, the scale or norm of the gradient becomes a secondary factor in terms of convergence. So, we can safely normalize each $M_i$ by the nuclear norm $\|M_i\|_* := \sum_{j=1}^{k_i} \sigma_j$ where $\sigma_j$ is the $j-$th singular value of $M_i$. This ensures that $I - M_i \succeq 0$, assistant convergence. While $\|M_i\|_*$ itself might be computationally expensive to compute, we may be able to use estimates of $\|M_i\|_*$ via simple procedures as in Qi (1984). In most practical implementations including ours, we simply compute the average of the gradients computed by each worker before sending it to the parameter server, that is, $k_i \equiv k = 1$ in which case simply normalizing by the euclidean norm is sufficient for our convergence result to hold. Our FA based distributed training Algorithm 1 solves the factored form for gradient aggregation purposes (in Step 6) at the parameter server.

Finally, please note that our technical assumptions are standard in optimization literature, that exploits smoothness of the objective function – since the feasible set of $Y$ in (1) is bounded, assumptions

are satisfied. Our proof techniques are standard, and we simply use them on our reformulation to obtain convergence guarantee **second order stationary points** for IRLS iterations since there exists a tractable SDP relaxation.

### E.2 FA Optimality Conditions and Similarities with Bulyan El Mhamdi et al. (2018) Baseline.

We first restate our Flag Aggregator with $g_i \in \mathbb{R}^{n \times k}$ in optimization terms as follows,

$$\min_{Y:Y^TY=I} A(Y) := \sum_{i=1}^{p} \sqrt{\left( 1 - \frac{\operatorname{tr}\left(Y^T g_i g_i^T Y\right)}{\operatorname{tr}\left(g_i^T g_i\right)} \right)} + \lambda \mathcal{R}(Y), \tag{18}$$

and write its associated Lagrangian $\mathcal{L}$ defined by,

$$\mathcal{L}(Y, \Gamma) := \sum_{i=1}^{p} \sqrt{\left( 1 - \frac{\operatorname{tr}\left(Y^T g_i g_i^T Y\right)}{\operatorname{tr}\left(g_i^T g_i\right)} \right)} + \lambda \mathcal{R}(Y) + \operatorname{tr}\left(\Gamma^T \left(Y^T Y - I\right)\right), \tag{19}$$

where $\Gamma \in \mathbb{R}^{m \times m}$ denotes the Lagrange multipliers associated with the orthogonality constraints in equation (18). In particular, since the constraints we have are equality, there are no sign restrictions on $\Gamma$, so they are often referred to as "free". Moreover, since $Y$ is a real matrix, the constraints are symmetric (i.e., $y_i^T y_j = y_j^T y_i$), we may assume that $\Gamma = \Gamma^T$, without loss of generality.

We will introduce some notations to make calculations easier. We will use $\tilde{g}_i \in \mathbb{R}^{n \times k}$ to denote the normalized gradients matrix of the data terms in equation (18). That is, we define

$$\tilde{g}_i := -\frac{1}{\operatorname{tr}\left(g_i^T g_i\right) \cdot \sqrt{\left( 1 - \frac{\operatorname{tr}\left(Y^T g_i g_i^T Y\right)}{\operatorname{tr}\left(g_i^T g_i\right)} \right)}} g_i g_i^T =: d_i g_i g_i^T. \tag{20}$$

With this notation, we are ready to use the first optimality conditions associated with the constrained optimization problem in (18) with its Lagrangian in (19) By first order optimality or KKT conditions, we have that,

$$0 = \nabla_Y \mathcal{L}(Y_*, \Gamma_*) = \left( \sum_{i=1}^{p} \tilde{g}_i \tilde{g}_i^T \right) Y_* + \lambda \nabla \mathcal{R}(Y_*) + 2Y_* \Gamma_*$$
$$= G D_* G^T Y_* + \lambda \nabla \mathcal{R}(Y_*) + 2Y_* \Gamma_*, \quad \text{(Objective)} \tag{21}$$
$$0 = \nabla_\Gamma \mathcal{L}(Y_*, \Gamma_*) = Y_*^T Y_* - I, \quad \text{(Feasibility)}$$

where $Y_* \in \mathbb{R}^{n \times m}, \Gamma_* \in \mathbb{R}^{m \times m}$ are the optimal primal parameters, lagrangian multipliers, and $D_* \in \mathbb{R}_{<0}^{p \times p}$ is the diagonal matrix with entries equal to $-d_i < 0$ as in equation (20). We may ignore the Feasibility conditions since our algorithm returns an orthogonal matrix by design, and focus on the Objective conditions. **Aside, IRLS corresponds to solving this lagrangian equations in (21) directly. When $\lambda = 0$, the solution to the lagrangian corresponds to computing the SVD of $GD$. While using functions that can be approximated as quadratic tr functions as regularization, $\nabla \mathcal{R}(Y) \equiv \mathcal{P}I$ where $\mathcal{P}$ is a positive semidefinite matrix, and so we have to compute the cholesky decomposition of $GDG^T + \lambda \mathcal{P}I$ first, and then a SVD as stated in Algorithm 1 in the main paper.**

Now, by bringing the term associated with Lagrangian to the other side, and then right multiplying by $\Gamma_*^{-1}$ inverse of $\Gamma_*$, we have that $Y_*$ satisfies the following identity,

$$Y_* = -\frac{1}{2} \left( G D_* G^T Y_* + \lambda \nabla \mathcal{R}(Y_*) \right) \Gamma_*^{-1}. \tag{22}$$

By using the identity (22), we can write an equivalent representation of our aggregation rule $Y_* Y_*^T G$ given by,

$$Y_* Y_*^T G = \frac{1}{4} \left( G D_* G^T Y_* + \lambda \nabla \mathcal{R}(Y_*) \right) \underbrace{\Gamma_*^{-1} \Gamma_*^{-1} \left( Y_*^T G D_* G^T + \lambda \nabla \mathcal{R}(Y_*)^T \right) G}_{:= \mathfrak{M}_* \in \mathbb{R}^{m \times p}}$$

$$\propto \left( G D_* G^T Y_* + \lambda \nabla \mathcal{R}(Y_*) \right) \mathfrak{M}_*$$

$$= G \underbrace{D_* G^T Y_*}_{:= S'_* \in \mathbb{R}^{p \times m}} \mathfrak{M}_* + \lambda \nabla \mathcal{R}(Y_*) \mathfrak{M}_*$$

$$= G S'_* \mathfrak{M}_* + \lambda \nabla \mathcal{R}(Y_*) \mathfrak{M}_*$$

$$:= G S_{\text{FA}} + \lambda \nabla \mathcal{R}(Y_*) \mathfrak{M}_*, \tag{23}$$

that is, the update rule of FA can be seen as a left multiplication with the square "flag selection" matrix $S_{\text{SA}} = S'_* \mathfrak{M}_* \in \mathbb{R}^{p \times p}$, and then perturbing with the gradient $\nabla \mathcal{R}(Y_*)$ of the regularization function $\mathcal{R}$ with a different matrix $\mathfrak{M}_*$ as in equation (23). Importantly, we can see in equation (23) that the (reduced) selection matrix $S \in \mathbb{R}_{\geq 0}^{p \times m}$ in Bulyan El Mhamdi et al. (2018) is equivalent to the total selection matrix $S_{\text{SA}} \in \mathbb{R}^{p \times p}$ in our FA setup. Moreover, we can also see that domain knowledge in terms of regularization function may also determine the optimal subspace, albeit additively only. We leave the algorithmic implications of our result as future work.

*Remark* E.3 (Invertibility of $\Gamma_*$ in Equation (22).). Theoretically, note that $\Gamma$ is symmetric, so by Spectral Theorem, we know that its eigen decomposition exists. So, we may use pseudo-inverse instead of its inverse. Computationally, given any primal solution $Y_*$ we can obtain $\Gamma_*$ by left multiplying equation (21) by $Y_*$ and use feasibility i.e., $Y_*^T Y_* = I$. Now, we obtain $\Gamma_*^{-1}$ columnwise by using some numerical solver such as conjugate gradient (with fixed iterations) on $\Gamma$ with standard basis vectors. In either case, our proof can be used with the preferred approximation choice of $\Gamma_*^{-1}$ to get the equivalence as in equation (23).

*Remark* E.4 (Provable Robustness Guarantees for FA.). Since our FA scheme is based on convexity, it is possible to show worst-case robustness guarantees for FA iterations under mild technical conditions on $Y^*$ – even under correlated noise, see for e.g. Assumption 1 in Klopp et al. (2016)). In fact, by using the selection matrix $S_{\text{FA}}$ in equation (23) in Lemma 1 in Damaskinos et al. (2019) and following the proof, we can get similar provable robustness guarantees for FA. We leave the theoretical analysis as future work.

# F ADDITIONAL EXPERIMENTS

## F.1 THE EFFECT OF REGULARIZATION PARAMETER

Our algorithm depends on the regularization parameter $\lambda$. Figure 10 below illustrates the effect of this parameter on similarity of aggregated gradient vectors for FA and Multi-Krum. For this experiment, we sample the gradients output by the parameter server across multiple epochs for both FA and Multi-Krum and compute the cosine similarity of corresponding vectors. We repeat the experiment with different $\lambda$ values. As we can see, for smaller iterations there is some similarity between the gradients computed by FA and Multi-Krum. This similarity is more visible for smaller $\lambda$ values.

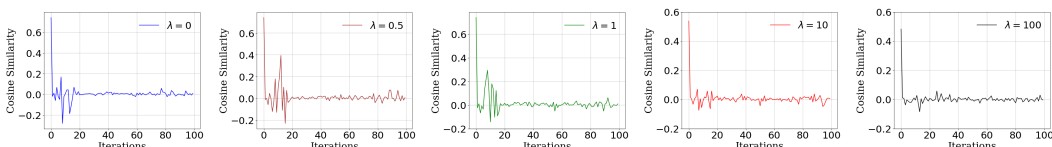

Figure 10: The effect of the regularization parameter $\lambda$ on similarity of FA performance to Multi-Krum

### F.2 EXPERIMENTS WITH OTHER BYZANTINE ATTACKS OR BASELINES

In Figure 11, we present a convergence plot to study FA when there are no Byzantine workers. Essentially, when $f = 0$, a robust aggregator should perform just as well as the standard distributed SGD with Mean as the aggregator. However, for $f > 0$, the Mean aggregator is not robust, as demonstrated in Figure 2. Figure 11 illustrates that FA outperforms the mean, which aligns with the findings of Mankovich et al. (2022).

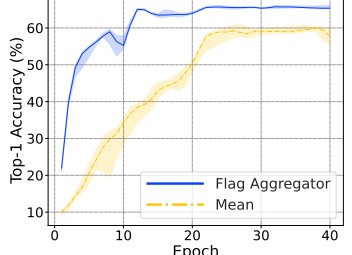

Figure 11: Convergence with $f = 0$

So far we have presented results where Byzantine workers send uniformly random gradient vectors, use synthetic data (nonlinear data augmentation routines), or where a percentage of gradients are dropped and zeroed out at the parameter server to show tolerance to communication loss. Here we provide more results when Byzantine workers send a gradient based on the Fall of Empires attack with $\epsilon = 0.1$ Xie et al. (2020) in Figure 12a and when they send 10x amplified sign-flipped gradients Allen-Zhu et al. (2021) in Figure 12b. Because mathematically, one iteration of FA with uniform weights assigned across all workers is equivalent to PCA, we also add a baseline for top-m principal components of the gradient matrix in Figure 12c. The novelty in our FA approach is the extension of PCA to an iteratively reweighted form that is guaranteed to converge. Specifically, we show that we obtain a convergent procedure in which we repeatedly solve weighted PCA problems. Moreover, the convergence guarantee immediately follows when the procedure is viewed as an IRLS procedure solving the MLE problem induced by the value of workers modeled with a beta distribution as in Section 2.2.

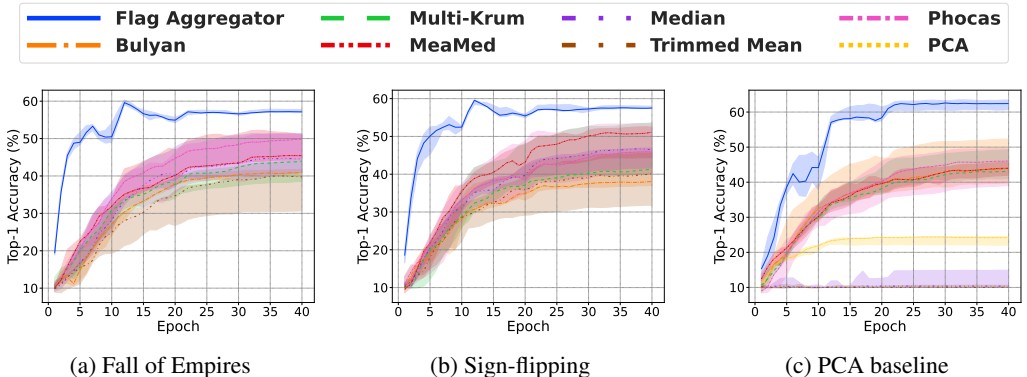

(a) Fall of Empires    (b) Sign-flipping    (c) PCA baseline

Figure 12: Robustness towards other attacks and comparison to PCA baseline, $p = 15, f = 2$.

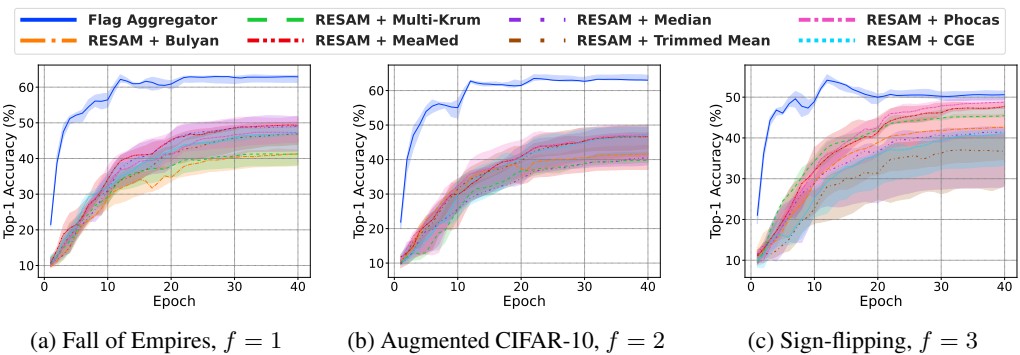

(a) Fall of Empires, $f = 1$    (b) Augmented CIFAR-10, $f = 2$    (c) Sign-flipping, $f = 3$

Figure 13: Robustness comparison with RESAM and CGE, $p = 15$.

We compared FA to RESAM Farhadkhani et al. (2022) that adapts the concept of gradient momentum to distributed architectures. RESAM involves each honest worker sending the momentums of their

stochastic gradients to the server, instead of just the gradients. Similar to Allouah et al. (2023a;b), RESAM still relies on resilient aggregation at the server, so we complement it with the existing baselines including Comparative Gradient Elimination (CGE) Gupta et al. (2021). CGE sorts the gradients based on their Euclidean norms and averages the gradients corresponding to the smallest $p - f$ norms. Figure 13 shows that FA outperforms all baselines under three different scenarios. In addition, FA is designed as a robust standalone aggregator that needs no extra computation or space for calculating gradient momentums and storing them on the worker side.

### F.3 EXPERIMENTS WITH THE TINY IMAGENET DATASET

We repeated our experiments with Tiny ImageNet Le and Yang (2015) which contains 100000 images of 200 classes (500 for each class) downsized to 64×64 colored images. We fix our batch size to 192 and use ResNet-50 He et al. (2016) throughout the experiments.

**Tolerance to the number of Byzantine workers:** In this experiment, we have $p = 15$ workers of which $f = 1, .., 3$ are Byzantine and send random gradients. The accuracy of test data for FA in comparison to other aggregators is shown in Figure 14. As we can see, for $f = 1$ and $f = 2$, FA converges at a higher accuracy than all other schemes. For all cases, FA also converges in ∼2x less number of iterations.

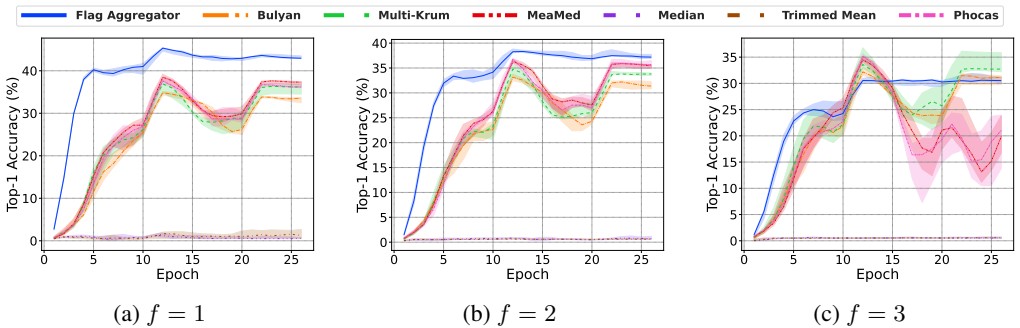

(a) $f = 1$        (b) $f = 2$        (c) $f = 3$

Figure 14: Tolerance to the number of Byzantine workers for robust aggregators.

**Tolerance to communication loss:** We set a 10% loss rate for the links connecting $f = 1, .., 3$ of the workers to the parameter server. Figure 15 shows that our takeaways in the main paper are also confirmed in this setting with the new dataset.

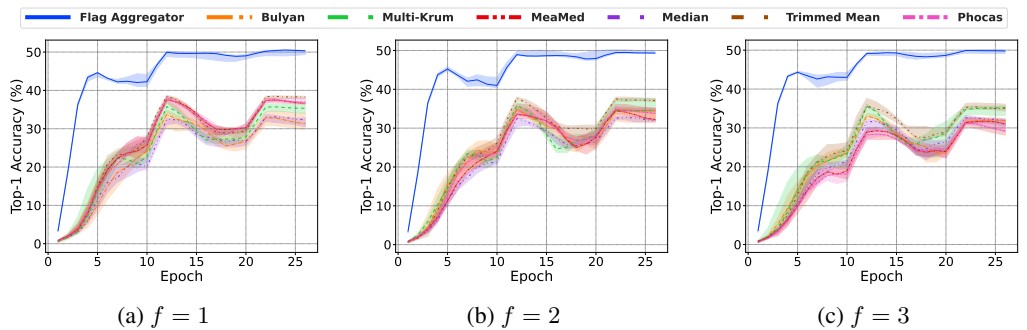

(a) $f = 1$        (b) $f = 2$        (c) $f = 3$

Figure 15: Tolerance to communication loss

**The effect of having augmented data during training in Byzantine workers:** As mentioned in the main paper, we choose two non linear augmentation schemes, Lotka Volterra (shown in rows 1 and 3 of Figure 16) and Arnold's Cat Map (shown in rows 2 and 4 of Figure 16). We used SciPy's Virtanen et al. (2020) solve_ivp method to solve the differential equations, by using the LSODA solver. In addition to the setup described in the main paper, we also added a varying level of Gaussian noise to each of the training images. All the images in the training set are randomly chosen to be augmented

with varying noise levels of the above mentioned augmentation schemes. We have provided the code that implements all our data augmentation schemes in the supplement zipped folder.

As seen from the figure, Arnold's Cat Map augmentations stretch the images and rearrange them within a unit square, thus resulting in streaky patterns. Whereas the Lotka Volterra augmentations distort the images while keeping the images similar to the original ones. We perform experiments with data augmented with varying shares using the two methods and show the results in Figure 17. For CIFAR-10, we showed the results when all of the samples in Byzantine workers are augmented in Figure 7 in the main paper. For Tiny ImageNet, this case is shown Figure 17a. Figures 17b and 17c show the results under different ratios on CIFAR-10. By changing the ratios we were interested to see if streaky patterns augmented by Arnold's Cat Map would introduce a more adverse effect from Byzantine workers compared to Lotka Volterra. Although the results do not show a significant signal, we can see that the augmentations did impact the overall gradients and that FA performs significantly better.

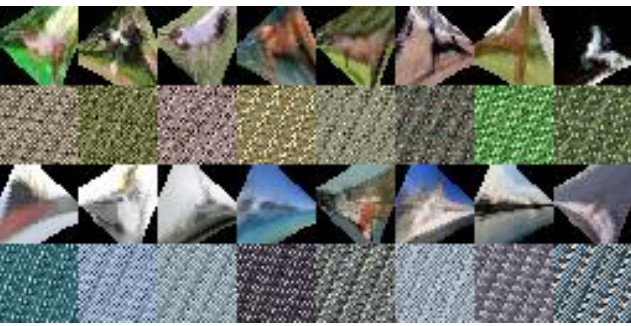

Figure 16: TinyImagenet data with Augmentation: **Row 1**: Lotka Volterra augmentation on Class Horse. **Row 2**: Arnold's Cat Map augmentation on Class Horse. **Row 3**:Lotka Volterra augmentation on Class Ship. **Row 4**: Arnold's Cat Map augmentation on Class Ship.

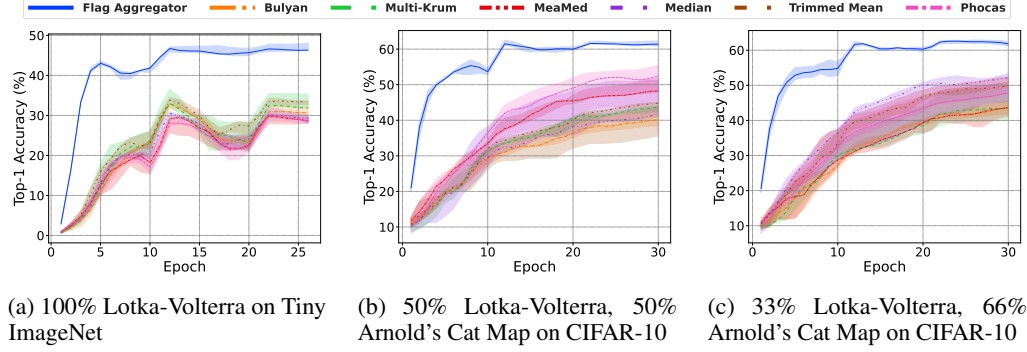

(a) 100% Lotka-Volterra on Tiny ImageNet

(b) 50% Lotka-Volterra, 50% Arnold's Cat Map on CIFAR-10

(c) 33% Lotka-Volterra, 66% Arnold's Cat Map on CIFAR-10

Figure 17: Accuracy of using augmented data in $f = 3$ workers

## G    DISCUSSION AND LIMITATION

**Is it possible to fully "offload" FA computation to switches?** Recent work propose that aggregation be performed entirely on network infrastructure to alleviate any communication bottleneck that may arise Sapio et al. (2021); Lao et al. (2021). However, to the best of our knowledge, switches that are in use today only allow limited computation to be performed on gradient $g_i$ as packets whenever they are transmitted Bosshart et al. (2013); McKeown (2015). That is, *programmability* is restrictive at the moment— switches used in practice have no floating point, or loop support, and are severely memory/state constrained. Fortunately, solutions seem near. For instance, Yuan et al. (2022) have already introduced support for floating point arithmetic in programmable switches. We may use quantization approaches for SVD calculation with some accuracy loss Song et al. (2018) to

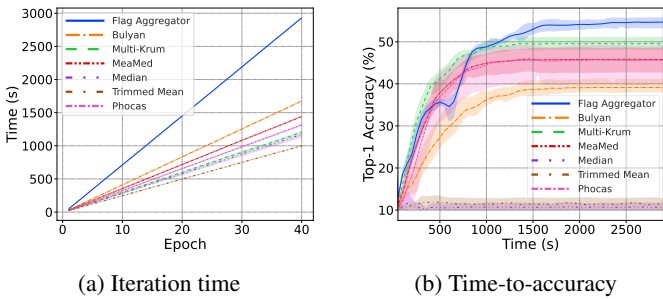

(a) Iteration time          (b) Time-to-accuracy

Figure 18: Wall clock time comparison

approximate floating point arithmetic. Offloading FA to switches has great potential in improving its computational complexity because the switch would perform as a high-throughput streaming parameter server to synchronize gradients over the network. Considering that FA's accuracy currently outperforms its competition in several experiments, an offloaded FA can reach their accuracy even faster or it could reach a higher accuracy in the same amount of time.

**Potential Limitation:** Since we perform SVD in every iteration of FA, the complexity of the algorithm would be $O(nN_\delta(\sum_{i=1}^p k_i)^2)$ with $N_\delta$ being the number of iterations for the algorithm. Figure 18 shows the wall clock time it takes for FA to reach a certain epoch (18a) or accuracy (18b) compared to other methods under a fixed amount of random noise $f = 3$ with $p = 15$ workers on CIFAR-10. Although the iteration complexity of FA is higher, each iteration has a higher utility as reflected in the time-to-accuracy measures. This makes FA comparable to others in a shorter time span, however, eventually FA converges to a better state as shown in Figure 18b. Regardless, it is possible to take advantage of fast, randomized SVD solvers to lower the wall clock time. In detail, to calculate the left singular values of $GD^{1/2} \in \mathbb{R}^{n \times p}$, we use the fact that number of workers $p \ll n$ and solve the $p \times p$ eigenvalue problem which can be fast in practice. Upon receiving the right singular vectors, first order methods can be used to obtain the left singular vectors. In this sense, we can use any fast, randomized SVD algorithm to solve for the right and/or left singular vectors.

**Towards Federated Learning Environments:** As we extend our gaze beyond the current implementation, the potential of FA within federated learning environments emerges as an intriguing frontier. Our scalability investigations, as illustrated in Figure 9, serve as a preliminary foundation for this exploration. We show FA's adaptability to more complex distributed settings, hinting at its viability across expanded cluster configurations.

Envisioning the implementation of FA in a federated context, we propose a hierarchical architectural model. This model involves gradient-computing workers transmitting their results to designated aggregating workers within clusters, akin to a parameter server but on a localized scale. These aggregating nodes then engage in further synthesis of results across multiple clusters, fostering a scalable, federated framework. This architecture not only maintains FA's core aggregation capabilities but also adapts them to the nuanced demands of federated learning, marking a significant stride toward broader applicability.

**Charting the Path Forward:** Our goal is to enhance the robustness and scalability of distributed training systems. By exploring the expansion of FA to more clusters and investigating computational offloading, we highlight our commitment to leading progress in the field. As we advance, our efforts are aimed at not just refining FA but also devising innovative solutions to meet the broad spectrum of challenges in distributed training.

