# OpenReview forum: "Flag Aggregator: Scalable Distributed Training under Failures and Augmented Losses using Convex Optimization"
_ICLR.cc/2024/Conference — ICLR 2024 poster_

### Official Review · Reviewer_Es7u · 2023-11-01

**Soundness:** 4 excellent
**Presentation:** 2 fair
**Contribution:** 3 good
**Rating:** 6
**Confidence:** 5

**Summary:**

This work proposes a novel robust aggregator, i.e., Flag Aggregator (FA), to deal with Byzantine faults in distributed computation. FA is an optimization-based subspace estimator that formulates aggregation as a maximum likelihood estimation procedure using Beta densities. In distributed training setups where vanilla mean aggregators are replaced by robust aggregators without additional tricks, FA consistently outperforms many other existing robust aggregators in extensive experiments with various batch sizes, fractions of Byzantine nodes, and number of nodes.

**Strengths:**

1. The proposed FA is novel in the distributed optimization literature. FA makes use of dependence among distributed gradients, while most existing aggregators only exploit the pairwise distance or moment conditions. This is beneficial in that if one can design aggregator that utilizes the data dependence, this aggregator is actually adaptive to this specific problem automatically, and thus one can improve performance. The authors provide insights on the intuitions, formulations, and approximate solutions to FA.
3. Consistent empirical performance improvement against many existing aggregators is achieved, among extensive experiments that investigate many facets of the Byzantine distribution optimization problem.

**Weaknesses:**

1. There are no specific examples that can show FA is theoretically better than existing methods, and thus makes the claim not persuasive to some audience. It can be, possibly, FA only performs well in the specific datasets, models, the way of distributing datasets among compute nodes in the experiments presented in this work. Even a naive toy example would help people understand why FA is theoretically better.
2. There are no comparisons between FA and existing aggregators in terms of computation complexity. It can be that, FA is too expensive to use in every iteration.
3. The intuition behind the development of FA is not clear enough to me, this goes to the first bullet, can the authors provide an example or a simple problem model to illustrate this? And why is FA optimal as claimed in line 83? Can the authors elaborate on this point?

**Questions:**

1. In line 22, the description of quadratic function is not clear to me, can the authors put it simpler?
2. How will the augmented data and stable diffusion influence the mutual dependence of distributed data, can the authors comment on that?
3. In line 72, what does it mean for noise to be nonlinear?
4. In line 73 & 74, how does the discrete hyper parameters hamper convergence of the overall training procedure, can you elaborate on that?
5. In line 76, the authors mention 'sparse', does that correspond to the choice of the second dimension of $Y$, i.e., $m$, how is $m$ chosen?
6. In line 115, it seems that $g$ should $g_i$, and there is no need to use trace on a scalar right?
7. In figure 3, I don't understand what is optimal subspace, do you mean optimality in the sense of formulation (5)?
8. Why is beta distribution used in the formulation of FA?
9. In the experiment starting from line 300, what attacks are being used?

**Details Of Ethics Concerns:**

N/A.

---

> ### Author Response · Authors · 2023-11-22
>
> **Q.** There are no specific examples that can show FA is theoretically better than existing methods, and thus makes the claim not persuasive to some audience. It can be, possibly, FA only performs well in the specific datasets, models, the way of distributing datasets among compute nodes in the experiments presented in this work. Even a naive toy example would help people understand why FA is theoretically better.
>
> **Ans.** Please see Figure 3 for a specific example. Here, we extracted gradients of parameters of ResNet-18 at $p=15$ workers, of which $f=2$ produce uniformly random gradients for 1000 iterations. At each iteration, we compute the mean of gradients and use it to calculate the values of workers. We also compute the optimal subspace provided by FA and calculate the values of workers using that at each iteration. The histogram of values computed using the mean gradient is much lower since most values are close to zero whereas when the optimal subspace provided by FA aggregation is used, the value is enhanced. We observed this phenomenon on many datasets, and architectures at various training stages, and we are happy to give more details with exact checkpoints in the Appendix for scientific replication purposes. In summary, we believe that this decreased value while using average gradients is due to the high dimensional phenomenon and can be easily replicated.
>
>
> We used the standard data-parallel training setup used in practice. First, we shuffle the data to randomize it, using a seed for consistency. Then, we create partitions proportional to the number of workers, ensuring each chunk is a different part of the dataset. These partitions are lists of indexes, representing how the data is divided. This setup allows distributing different data chunks to multiple workers in a training process, ensuring each worker gets a unique, random subset of the data for independent and identically distributed gradients.
>
> We assure the reviewer that the way of distributing datasets did not have any effect on our experiments. We consider the reproducibility aspects of our experiments seriously, so we will release all the datasets, along with code, model, a blog post, as well as a tutorial to reproduce all the figures in our submission on acceptance.
>
> **Q.** There are no comparisons between FA and existing aggregators in terms of computation complexity. It can be that, FA is too expensive to use in every iteration.
>
> **Ans.** Please refer to the potential limitation in the appendix and the corresponding Figure~18. Please note that although the iteration complexity of FA is higher, the utility of each iteration is higher, which is reflected in the time-to-accuracy comparison. The figure shows two main points: 1) FA converges to a higher accuracy, and more importantly, 2) it reaches to that level in less (wall-clock) time. In this sense, we think that FA's somewhat higher per-iteration complexity is well justified. In the future, we plan on evaluating eigenvalue solvers to further accelerate training time. We have provided a more detailed explanation, along with preliminary evaluation of per-iteration complexity in Section G in the Appendix. In essence, we believe that this gap in per-iteration time can be further reduced using modern fast eigenvalue solvers.
>
> **Q.** The intuition behind the development of FA is not clear enough to me, this goes to the first bullet, can the authors provide an example or a simple problem model to illustrate this? And why is FA optimal as claimed in line 83? Can the authors elaborate on this point?
>
> **Ans.** FA is optimal in the sense that it seeks to find an approximate solution to the optimization problem formulation given in Equation (5). Our derivation using MLE in Sec 2.2 shows that problem (5) can be used whenever the value of the gradients needs to be maximized or in other words, when we want to maximize the values provided by gradients from workers in a distributed training pipeline. Finally, to see that FA aggregation guarantees convergence, please notice that our update is given by $YY^{\top}G{\bf 1}$ where $G{\bf 1}$ simply denotes the sum of gradients from workers. Now since $YY^{\top}$ is a symmetric positive semidefinite matrix, it is guaranteed to have an eigenbasis. Intuitively, this means that the update rule itself simply rotates the gradients, scales them in specific coordinates, and rotates them back to their original basis. This is our main intuition in proposing subspace based aggregation that is currently not explored in distributed training pipelines used in practice.

---

> > ### Author Response · Authors · 2023-11-22
> >
> > **Q.** In line 22, the description of quadratic function is not clear to me, can the authors put it simpler?
> >
> > **Ans.** We mean that $A_{g_1,\cdots,g_p}(Y)=\sum_{i=1}^p\\|Y-g_i\\|_2^2$ where $Y\in\mathbb{R}^{n\times 1}$
> >  (or simply $C=\mathbb{R}^n$) where $n$ is the number of training parameters.
> >
> > **Q.** How will the augmented data and stable diffusion influence the mutual dependence of distributed data, can the authors comment on that?
> >
> > **Ans.** Thank you for the question! When data augmentation is independent of the training data set, then the mutual independence in data obtained by augmentation is preserved since functions of independent variables are independent, so no dependence after augmentation. However, when models such as stable diffusion or latent space or feature based augmentations are used, independence is not preserved anymore. This is because parameters of stable diffusion, encoders used in obtaining latent representations, features are trained using training data set -- information from individual data points in the training data set can influence the parameters of the models.  So we cannot assume independence of augmented data using these models, and dependence increases.
> >
> > **Q.** In line 72, what does it mean for noise to be nonlinear?
> >
> > **Ans.** If $g$ is the true gradient, then noise $\epsilon$ is *linear* if the obtained or observed gradient can be written as $g+\epsilon$. If the obtained gradient cannot be decomposed in this way, we call the noise to be nonlinear. The same condition can be used for training data samples $x$. In our submission, we mention two specific nonlinear noise models in Sec 3.1 near L283-290 viz., Lotka-Volterra and Arnold Cat Map, for imaging datasets.
> >
> > **Q.** In line 73 \& 74, how does the discrete hyper parameters hamper convergence of the overall training procedure, can you elaborate on that?
> >
> > **Ans.** For every value of discrete hyperparameter, the training usually needs to be restarted from scratch, and so we may require multiple training runs to choose the optimal value.
> >
> > **Q.** In line 76, the authors mention 'sparse', does that correspond to the choice of the second dimension of $Y$, i.e., $m$, how is $m$ chosen?
> >
> > **Ans.** Yes. We believe that the reviewer is pointing out this is a discrete parameter in our setting. Thank you, we completely agree with this subtle observation. Fortunately, while using FA, there is an ordering in our setting due to orthogonality constraints as explained in L135-148, so a higher $m$ can be used in practice. In this sense, it is advantageous to use FA compared to discrete parameters that are used in $k$-NN based algorithms. Experimentally, we observed that choosing a larger $m$ in the definition of $Y$ increases the wall clock time since a larger SVD has to be computed at every iteration but *not* the number of iterations for convergence.
> >
> > **Q.** In line 115, it seems that $g$ should be $g_i$, and there is no need to use trace on a scalar right?
> >
> > **Ans.** Yes, $g$ should be $g_i$, thank you for pointing. We have fixed the typo. The $\text{tr}(\cdot)$ is used in tractability analysis of our formulation in Equation (5). So we thought that it would be nice to hint to the reader of further developments in our submission. We are happy to implement suggestions from the reviewer to improve the readability of our submission.
> >
> > **Q.** In figure 3, I don't understand what is optimal subspace, do you mean optimality in the sense of formulation (5)?
> >
> > **Ans.** Yes. We have answered this in detail in your previous question asking why is FA optimal. We will include our explanation and any other suggestions from the reviewer.
> >
> > **Q.** Why is beta distribution used in the formulation of FA?
> >
> > **Ans.** In general, Beta distributions can be used to model events which are constrained to take place within an interval defined by a minimum and maximum value or bounded sets. Since the value of a worker, $v_i$ is in the unit interval $[0,1]$ by definition,  assuming values to be sampled from a Beta distribution is valid. Moreover,  parameters $\alpha,\beta$ used in the definition of Beta distribution can be used to indicate different mathematical quantities such as convexity, dispersion of the distribution of values in our formulation (5). We believe that the additional flexibility can enable faster training procedures (in terms of iteration complexity) which may be important in the distributed setups.
> >
> > **Q.** In the experiment starting from line 300, what attacks are being used?
> >
> > **Ans.** In this experiment, $f$ workers produce uniformly random gradients in $[0, 1]$ so has coordinatewise mean  is $0.5$ and standard deviation is $ \sqrt{\frac{1}{12}}$.

---

> > ### Comment · Reviewer_Es7u · 2023-11-23
> >
> > I express my gratitude to the reviewers for providing thorough responses to my inquiries. While I acknowledge that the suggested aggregator consistently demonstrates strong performance across diverse datasets and under various attacks, my initial concern remains regarding the absence of a theoretical toy example. This example would serve to illustrate: 1) the modeling process, elucidating the rationale behind determining the appropriate value for $m$; and 2) a comparison of computational complexity with other methods, including the intricacies of solving FA. I understand that this may not be deemed necessary if the primary emphasis of the contribution is on practical effectiveness, and raised my scores accordingly.

---

### Official Review · Reviewer_PqPx · 2023-11-01

**Soundness:** 2 fair
**Presentation:** 3 good
**Contribution:** 2 fair
**Rating:** 6
**Confidence:** 2

**Summary:**

This work proposes an aggregation function based on low-rank projection. In particular, for a given matrix $G$, this work proposes to perform the aggregation by projecting it to a low-dimensional space, $YY^TG1$, where $Y$ is the subspace chosen based on low-rank factorization of $G$. The connection between the algorithm and the Maximum Likelihood Estimation procedure using Beta densities is presented. Experimental results show improvements in communication efficiency and accuracy compared to previous works.

**Strengths:**

1. The proposed method is simple yet effective. Theoretical analysis is also provided to backup the algorithm.

2. Detailed experimental results are provided, and better accuracy has been shown compared to the previous works.

**Weaknesses:**

Here are a few comments regarding the experimental section:

1. The choice of setting the subspace rank $m$ to $(p+1)/2$ in all experiments raises a question. Is this decision rooted in theoretical analysis or other considerations?

2. The paper introduces a general framework in Section 2, considering a general norm and suggesting SDP for solving the system. However, the focus shifts to $\ell_1$ regularization later due to its ease of optimization. An experimental comparison of different regularization terms could bridge the apparent gap between Sections 2 and 3.

3. It would be useful to explore whether the proposed algorithm offers advantages even when $f=0$.

**Questions:**

n/a

---

> ### Author Response · Authors · 2023-11-22
>
> **Q.** Is the choice of setting the subspace rank $m$ to $(p+1)/2$ in all experiments rooted in theoretical analysis or other considerations?
>
> **Ans.** Great question! We decided to use $m$ to be $(p+1)/2$ since it corresponds to a naive prior that each worker may provide noisy gradient with probability $1/2$. In this case, the expected number of clean gradients will be the integer greater than or equal to  $p/2$.  Our choice also corresponds to a realistic setting in most use cases -- that the majority (i.e., more than half) of workers provide correct gradients.
>
> **Q.** The paper introduces a general framework in Section 2, considering a general norm and suggesting SDP for solving the system. However, the focus shifts to regularization later due to its ease of optimization. An experimental comparison of different regularization terms could bridge the apparent gap between Sections 2 and 3.
>
> **Ans.** We use the SDP relaxation of the MLE in Equation (4) to argue that solving FA problem may be tractable since SDPs can be solved efficiently from the theoretical standpoint. We provided the regularized version with two specific regularization functions $\mathcal{R}$ that our tractability analysis using SDP can handle since these two choices of $\mathcal{R}$ are convex functions with respect to $Y$. We provided preliminary experiments with pairwise data dependent regularizer in Sec F.2, Figure 10. We plan to do a detailed study on different regularizers with FA in the near future. We will definitely clarify this in the camera-ready.
>
> **Q.** It would be useful to explore whether the proposed algorithm offers advantages even when $f=0$
>
>
> **Ans.** We agree, thank you for pointing this out!  Our update rule is given by $YY^{\top}G{\bf 1}$ where $G{\bf 1}$ simply denotes the sum of gradients from workers. Reviewer will be able to see that since $YY^{\top}$ is a symmetric positive semidefinite matrix, it is guaranteed to have an eigenbasis. Intuitively, this means that the update rule itself simply rotates the gradients, scales them in specific coordinates, and rotates them back to their original basis. So, we may think of $YY^{\top}$ as an adaptive preconditioner and can be used whenever the loss landscape is ill-conditioned. In the Appendix, we have added a convergence plot in Figure 11 for the noiseless setting i,e, $f=0$. In this setting, a robust aggregator should be as good as the standard distributed SGD with Mean as the aggregator, which we can clearly observe here.  In Figure 11 we can see that FA, performs better than mean, which is consistent with the findings in [Mankovich et al. 2022].
>
> [Mankovich et al. 2022] The Flag Median and FlagIRLS, CVPR 2022.

---

### Official Review · Reviewer_YEpb · 2023-11-05

**Soundness:** 3 good
**Presentation:** 3 good
**Contribution:** 3 good
**Rating:** 6
**Confidence:** 3

**Summary:**

This paper proposes Flag Aggregator, a simple Maximum Likelihood Based estimation procedure for aggregation purposes. They show that any procedure used to solve Flag Optimization can be directly used to obtain the optimal summary statistic $Y^*$.The authors also show the approach is resilient against Byzantine attacks for gradient aggregation.

**Strengths:**

- Gradient aggregation is a critical design choice in many of the distributed training applications, and is ubiquitous. The proposed method seems promising and useful for this space.

**Weaknesses:**

- I am not sure how much overhead the SVD might bring in practice, could you provide some real-world measurement? So far all the empirical results are epoch-wise measuring.
- Most of the baseline compared in the experiments seem to be from at least five years ago (2018); I wonder if the authors can compare their approach with latest algorithms? For instance  Allouah et al. (2023a;b); Farhadkhani et al. (2022) as mentioned in the related work.

**Questions:**

I'll increase my rating if comparison to more recent algorithms is provided.

---

> ### Author Response · Authors · 2023-11-22
>
> **Q.** How much overhead the SVD might bring in practice, could you provide some real-world measurement?
>
> **Ans.** Yes, please refer to the potential limitation in Section G in the appendix. Indeed, there is nontrivial overhead to execute FA in each training iteration. Fortunately, this overhead is compensated by reduction in total number of training iterations in all our experiments, thus providing significantly lesser wall clock time. We have provided more explanations and suggestions for potentially accelerating training time by using approximate eigenvalue calculation. When using exact eigenvalue calculations, the overheads per iteration can be seen in Figure 18.
>
> **Q.** Most of the baseline compared in the experiments seem to be from at least five years ago (2018). Can the authors compare their approach with latest algorithms? For instance Allouah et al. (2023a;b); Farhadkhani et al. (2022)
>
> **Ans.** Thank you for the suggestion. We compared FA to RESAM [Farhadkhani et al. (2022)] which adapts the concept of gradient momentum to distributed architectures. RESAM involves each honest worker computing and storing the momentums of their stochastic gradients as well as the gradients, and sending the momentum to the server which increases computational complexity per iteration. Similar to [Allouah et al. (2023a;b)], RESAM still relies on resilient aggregation similar to our baselines at the server, so we complement it with the existing baselines including CGE [Gupta et al. (2021)]. Nevertheless, we show the comparison in Figure~13 of the updated submission while ignoring the overheads of RESAM. The figure shows FA's advantage under three different scenarios. In addition, FA is designed as a robust standalone aggregator that needs no extra computation or space for calculating gradient momentums and storing them on the worker side.
>
> [Gupta et al. (2021)] Byzantine Fault-Tolerant Distributed Machine Learning Using Stochastic Gradient Descent (SGD) and Norm-Based Comparative Gradient Elimination (CGE)
>
> [Farhadkhani et al. (2022)] Byzantine machine learning made easy by resilient averaging of momentum
>
> [Allouah et al. (2023a)] Distributed Learning with Curious and Adversarial Machines
>
> [Allouah et al. (2023b)] Fixing by Mixing: A Recipe for Optimal Byzantine ML under Heterogeneity

---

### Author Response · Authors · 2023-11-22

Dear Reviewers,

We are grateful for your time, and for your excellent comments. We have answered all questions and addressed concerns brought up in the individual reviews. In the revised submission, we have included new experiments with very recent robust training algorithms and figures for noiseless settings. Reviewer Es7u has many minor technical questions that we have answered to the fullest extent possible. We hope that this leads to even stronger and enthusiastic support for our paper.

---

### Meta-Review · Area_Chair_YRGW · 2023-12-08

**Metareview:**

This paper introduces a novel approach to robust aggregation for distributed training with Byzantine failures. The approach is based on Flag Aggregation, which poses the aggregation problem as a MLE with Beta variables. Experiments illustrate the promise of the approach for distributed training across various batch sizes, various fractions of Byzantine nodes, and network sizes.

All reviewers agree that the proposed approach is novel. The method is simple but well-motivated from first principles, and that the experiments convincingly demonstrate the promise of the approach.

Some concerns were raised about comparisons with recent baselines, certain choices made in the empirical comparison, and providing more intuition about the approach. Some of these were addressed adequately during the rebuttal.

The paper could still be strengthened by adding an additional intuitive example.

**Justification For Why Not Higher Score:**

I wouldn't be opposed to seeing this paper receive a spotlight, but I'm not sure if the reviewer scores/feedback justify it.

**Justification For Why Not Lower Score:**

The approach is novel, with an extensive empirical evaluation. All reviewers agree that this is a solid contribution, and there has been sustained interest in dealing with Byzantine failures in distributed training over several years. Given that this paper makes a strong contribution to the existing literature, I believe it is justified to accept it to the conference.

---

### Decision · Program_Chairs · 2024-01-16

Accept (poster)